# L1 retrotransposons exploit RNA m$^6$A modification as an evolutionary driving force

Sung-Yeon Hwang[1,2], Hyunchul Jung[3], Seyoung Mun[4,5,6], Sungwon Lee[1,2], Kiwon Park[1,2], S. Chan Baek[1,2], Hyungseok C. Moon[7], Hyewon Kim[1,2], Baekgyu Kim[1,2], Yongkuk Choi[1,2], Young-Hyun Go[8], Wanxiangfu Tang[9], Jongsu Choi [10], Jung Kyoon Choi [3], Hyuk-Jin Cha [8], Hye Yoon Park [7], Ping Liang [9], V. Narry Kim [1,2], Kyudong Han[5,6,11✉] & Kwangseog Ahn[1,2✉]

L1 retrotransposons can pose a threat to genome integrity. The host has evolved to restrict L1 replication. However, mechanisms underlying L1 propagation out of the host surveillance remains unclear. Here, we propose an evolutionary survival strategy of L1, which exploits RNA m$^6$A modification. We discover that m$^6$A 'writer' METTL3 facilitates L1 retrotransposition, whereas m$^6$A 'eraser' ALKBH5 suppresses it. The essential m$^6$A cluster that is located on L1 5′ UTR serves as a docking site for eukaryotic initiation factor 3 (eIF3), enhances translational efficiency and promotes the formation of L1 ribonucleoprotein. Furthermore, through the comparative analysis of human- and primate-specific L1 lineages, we find that the most functional m$^6$A motif-containing L1s have been positively selected and became a distinctive feature of evolutionarily young L1s. Thus, our findings demonstrate that L1 retrotransposons hijack the RNA m$^6$A modification system for their successful replication.

[1] Center for RNA Research, Institute for Basic Science, Seoul, Republic of Korea. [2] School of Biological Sciences, Seoul National University, Seoul, Republic of Korea. [3] Department of Bio and Brain Engineering, KAIST, Daejeon, Republic of Korea. [4] Department of Nanobiomedical Science, Dankook University, Cheonan, Republic of Korea. [5] DKU-Theragen institute for NGS analysis (DTiNa), Cheonan, Republic of Korea. [6] Center for Bio Medical Engineering Core Facility, Dankook University, Cheonan, Republic of Korea. [7] Department of Physics and Astronomy, Seoul National University, Seoul, Republic of Korea. [8] Department of Pharmacy, Seoul National University, Seoul, Republic of Korea. [9] Department of Biological Sciences, Brock University, St. Catharines, ON, Canada. [10] Dr. von Hauner Children's Hospital, Ludwig-Maximilians-University of Munich, Munich, Germany. [11] Department of Microbiology, Dankook University, Cheonan, Republic of Korea. ✉email: kyudong.han@gmail.com; ksahn@snu.ac.kr

Long interspersed element-1 (L1) is currently an active autonomous retrotransposon, and constitutes ~17% of the human genome[1]. The average human genome contains 80–100 copies of retrotransposition-competent L1s[2,3]. A retrotransposition-competent L1 is 6 kb in length and consists of a 5′ untranslated region (UTR) containing an internal promoter[4], two open reading frames (ORF1 and ORF2), and a short 3′ UTR. ORF1 encodes a nucleic acid chaperon protein (ORF1p)[5], while ORF2 encodes a protein with endonuclease and reverse transcriptase activity (ORF2p)[6,7]. ORF1p and ORF2p associate preferentially with their parental mRNA to form an L1 ribonucleoprotein (RNP) particle[8]. The L1 RNP enters the nucleus and then generate the progeny through de novo insertion of its cDNA[9,10]. The mobility of L1s contributed to a source of genetic variation, but also pose a threat to genome integrity[11–13]. Although several host factors have evolved to suppress L1 retrotransposition, the youngest L1 subfamilies are still active and replicated continuously[14–16]. To date, however, the mechanism of how L1s have propagated under host surveillance remains unknown.

N6-methyladenosine (m6A) is the most prevalent internal modification in eukaryotic mRNAs, which determines RNA function and fate[17]. Several enzymes dynamically process the m6A modification of mRNA. The methyltransferase-like enzyme METTL3, which is the catalytic subunit of the RNA methyltransferase complex, adds m6A at the consensus motif DRAmCH (where D = G/A/U, R = G/A, and H = U/C/A)[18,19]. Conversely, m6A is removed by the demethylases α-ketoglutarate-dependent dioxygenase AlkB homolog 5 (ALKBH5) or fat mass and obesity-associated protein (FTO)[20,21]. Emerging studies have revealed that m6A modifications in viral transcripts affect the gene expression and replication of viruses such as HIV-1[22]. Despite the critical role of m6A in pathogenic viral transcripts, it remains unclear whether m6A participates in the regulation of the endogenous parasites, L1 retrotransposons.

Here, we show that L1 retrotransposon exploits m6A modification to facilitate its mobility. We figured out that m6A machinery plays a role in L1 regulation and identified the functional m6A cluster located on 5′ UTR of retrotransposition-competent full-length L1. Our results show that L1 5′ UTR m6A cluster recruits eukaryotic initiation factor 3 (eIF3) for efficient translation and promotes the formation of L1 RNP, which are essential for L1 mobility. Lastly, we traced a recent episode of human- and primate-specific L1 evolution and revealed that the most functional m6A site (A332 residue in L1 5′ UTR) first appeared ~12 million years ago. During the primate evolution, A332 m6A-positive L1s have been selected and became a distinctive feature of evolutionarily young L1s, which suggests that the acquisition of m6A motif has acted as an evolutionary driving force for L1 retrotransposons.

## Results

**METTL3 and ALKBH5 regulate L1 retrotransposition.** To determine whether RNA m6A modification affects L1 retrotransposition, we evaluated the effects of the RNA m6A machinery on L1 retrotransposition using a cell-based engineered L1-reporter assay[23]. For the assay, we used the pJJ101-L1-dn6 2.2 construct (hereafter referred to as pL1Hs) that contains a blasticidin S deaminase gene (*mblastI*) within the 3′ UTR antisense to the SV40 promoter[24,25] (Fig. 1a). When L1 is successfully integrated into the host chromosome, the cells acquire resistance to blasticidin (Fig. 1a).

We depleted the m6A methyltransferase METTL3, RNA demethylase ALKBH5, and FTO using small-interfering RNA (siRNAs) in HeLa cells and transfected pL1Hs vector. In METTL3-depleted cells, the number of blasticidin S-resistant colonies, which represent successful L1 retrotransposition, was reduced by >2-fold compared to that of control siRNA (Fig. 1b and Fig. S1a). Conversely, the silencing of ALKBH5 increased L1 mobility, while the silencing of FTO did not affect L1 retrotransposition (Fig. 1b). The depletion of the m6A machinery did not vitiate cell viability (Fig. S1b). In a reciprocal experiment, we performed an L1 retrotransposition assay with the ectopic expression of RNA m6A demethylase ALKBH5 or FTO. Notably, the overexpression of ALKBH5 inhibited L1 mobility by ~4-fold, whereas FTO overexpression did not affect L1 mobility compared to that in AcGFP-expressing negative control cells (Fig. 1c and Fig. S1c). We hypothesized that ALKBH5 may function as an L1 restriction factor by removing essential m6A for L1 mobility. To examine whether the enzymatic function of ALKBH5 is critical for L1 mobility suppression, we performed L1 assays using the plasmid-encoding catalytically inactive mutant of ALKBH5 (ALKBH5H204A). As anticipated, ALKBH5 could successfully restrained L1 mobility to levels that were comparable to that suppressed by a reverse transcription inhibitor (stavudine; d4T), whereas ALKBH5H204A overexpression did not result in the restriction of L1 mobility (Fig. 1d and Fig. S1d). The viability of transfected cells remained unaffected (Fig. S1e).

The pL1Hs plasmid encodes reporter L1 downstream of the CMV promoter and L1 5′ UTR promoter. Since the presence of the CMV promoter might affect L1-associated m6A modification, we used a pYX014 L1-luciferase vector driven only by the L1 5′ UTR promoter. Using pYX014, the firefly luciferase reporter within the 3′ UTR allowed us to assess L1 mobility by measuring luminescence as previously reported[26] (Fig. S1f). Overexpression of ALKBH5 impaired L1 retrotransposition, regardless of the presence of the CMV promoter (Fig. S1g). In-line with this result, depletion of METTL3 or ALKBH5 regulates L1 mobility, whereas FTO knockdown did not affect (Fig. S1h). These results indicate that ALKBH5-specific m6A substrates are necessary for L1 expansion. To summarize, our data support the functional role of the m6A machinery in regulating L1 retrotransposition.

RNA m6A metabolism regulates gene expression at post-transcriptional levels. Therefore, we speculated that the m6A machinery would influence the protein expression of L1. Immunoblot analysis of HeLa cells devoid of m6A enzymes revealed that m6A enzymes regulate the expression of ORF1p (Fig. 1e). Overexpression of ALKBH5 inhibited ORF1p expression, while the ectopic overexpression of FTO and ALKBH5H204A did not affect the ORF1p expression (Fig. 1f, g). In each condition, the transfection efficiency of pL1Hs was not affected by siRNA or plasmids transfection (Fig. S2a, b). Furthermore, neither the depletion of RNA m6A machinery nor the over-expression of ALKBH5 altered the levels of expression of the control EGFP (Fig. S2c, d), which indicates that m6A enzymes do not affect transfection efficiency. These results suggest that m6A-mediated L1 regulation affects both retrotransposition and L1 protein expression.

**L1 RNA is modified by m6A.** Although the possibility of L1 m6A modification was demonstrated in recent studies[27,28], it remains unclear whether m6A modification occurs in retrotransposition-competent full-length L1, and if so, which region of the L1 transcript is modified by m6A. To validate whether m6A modifies L1 RNA, we performed methyl-RNA immunoprecipitation (MeRIP) using human embryonic stem cells (H9 hESCs) that express endogenous L1 at sufficient levels[29,30]. Through qRT-PCR analysis of the MeRIP eluates, we detected the enrichment of L1 RNA at a level comparable to that for known m6A-modified *SON* and *CREBBP* mRNA, but

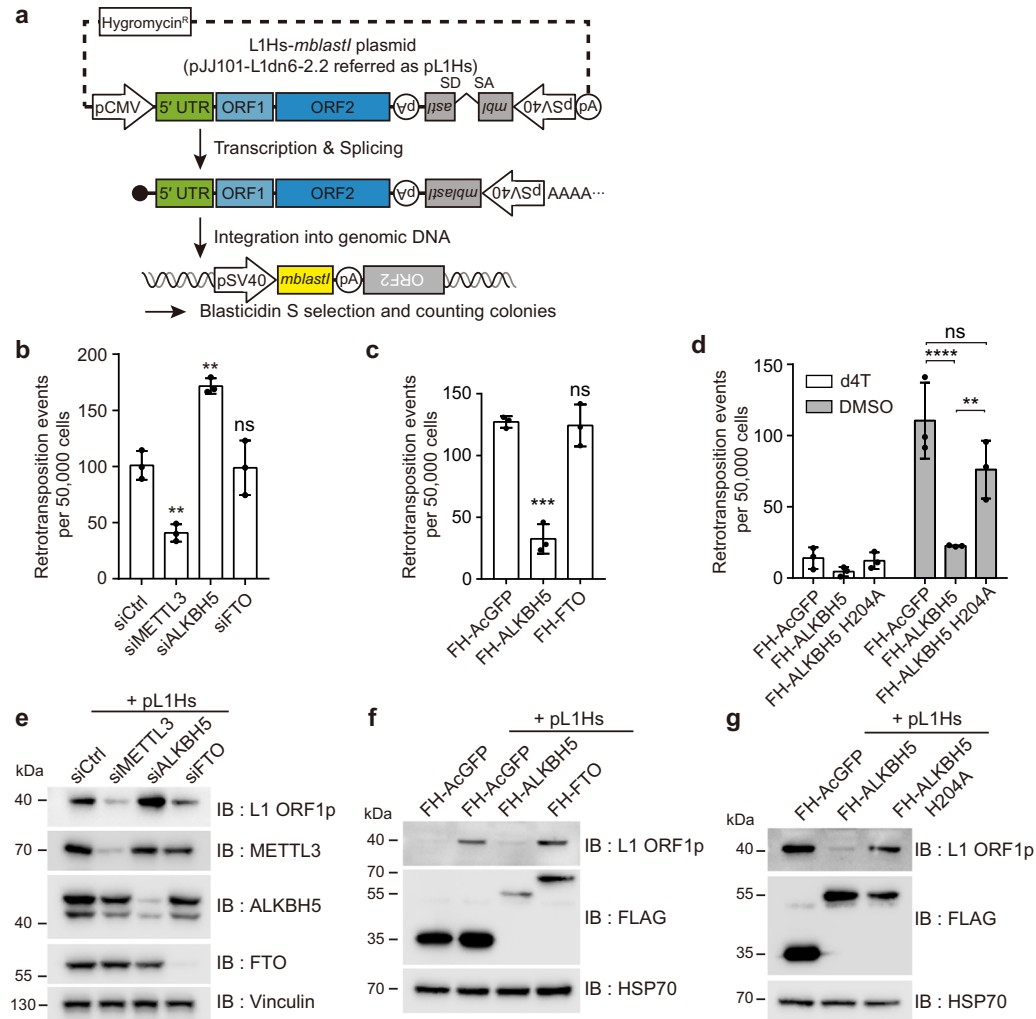

**Fig. 1 RNA methylation machinery controls L1 retrotransposition. a** A schematic of the L1 construct and an overview of the L1 retrotransposition assay using engineered human L1 construct. **b** Retrotransposition assay in HeLa cells treated with siRNA that targets METTL3, ALKBH5, or FTO. A nontargeting siRNA (siCtrl) was used as a control. **c** Retrotransposition assays performed by co-transfecting the pL1Hs expression cassette with the indicated $m^6A$ enzyme-expressing vectors into HeLa cells. **d** L1 retrotransposition assays were performed in ALKBH5, ALKBH5 catalytically inactive mutant (H204A), or AcGFP(control)-overexpressing cells. Cells treated with 50 μM stavudine (d4T) served as a reverse transcription negative control. ($n = 3$ independent samples, mean ± s.d., one-way ANOVA and Tukey's multiple comparisons test; $****p < 0.0001$, $***p < 0.001$, $**p < 0.01$, in comparison to control, ns: not significant). **e** Immunoblot assay of lysates from pL1Hs-transfected HeLa cells treated with indicated siRNAs that target $m^6A$ enzymes. Vinculin served as a loading control. **f**, **g** Immunoblot assay using pL1Hs-expressing HeLa cells. AcGFP, ALKBH5, FTO, or ALKBH5$^{H204A}$ overexpression plasmids were co-transfected with pL1Hs. FH-AcGFP served as transfection control. HSP70 served as a loading control. The predicted molecular weight of FLAG-HA-tagged proteins are 34 kDa for FH-AcGFP, 51 kDa for FH-ALKBH5, and 65 kDa for FH-FTO. The immunoblot images (**e**–**g**) are representative of three independent experiments. Source data are provided as a Source data file.

much more than negative control *HPRT1* mRNA (Fig. 2a). To minimize bias resulting from primers in L1 RNA detection, we used three different primer sets that targeted the 5′ UTR, ORF1, and ORF2 regions and did not observe significant differences in the results obtained for these primers (Fig. 2a). Similar to the endogenously expressed L1 RNA in hESCs, MeRIP-qPCR analysis clearly demonstrated that the L1 RNA exogenously expressed in HeLa cells undergoes $m^6A$ modification (Fig. S3a). We then evaluated if the silencing of METTL3 or ALKBH5 would alter the extent of $m^6A$ modification of the L1 RNA. Indeed, MeRIP-qPCR with METTL3-depleted cells revealed lower enrichment of the $m^6A$-modified L1 than of siCtrl-treated cells, whereas ALKBH5 knockdown augmented the levels of $m^6A$-positive L1 (Fig. 2b). These results indicate that METTL3 can install $m^6A$ modification in L1 transcripts, while ALKBH5 plays a role in removing the modification.

To examine the $m^6A$-modified regions in the L1 transcripts, we analyzed the $m^6A$ transcriptome of hESCs reported previously[31] and mapped reads to the consensus sequence of L1Hs, the youngest L1[32]. We identified 18 peaks across the L1Hs sequence using two biological replicates (Fig. 2c). Given that the reads from L1s may yield false-positive results, we narrowed down and selected the peaks that are likely to contain $m^6A$ motifs from 18 peaks through the $m^6A$ prediction score algorithm (SRAMP)[33]. SRAMP analysis revealed that the 9 peaks found in the ORF1, ORF2, and 3′ UTR regions do not contain $m^6A$ motifs, and that only the 6 peaks located at 5′ UTR have potential $m^6A$ motifs (Table S1a).

Next, we mapped the sites of $m^6A$ modifications in reporter L1-transfected HeLa cells using MeRIP-seq. Consistent with findings from previous studies[18,19], our results indicated that the transcriptome-wide distribution of $m^6A$ peaks were preferentially

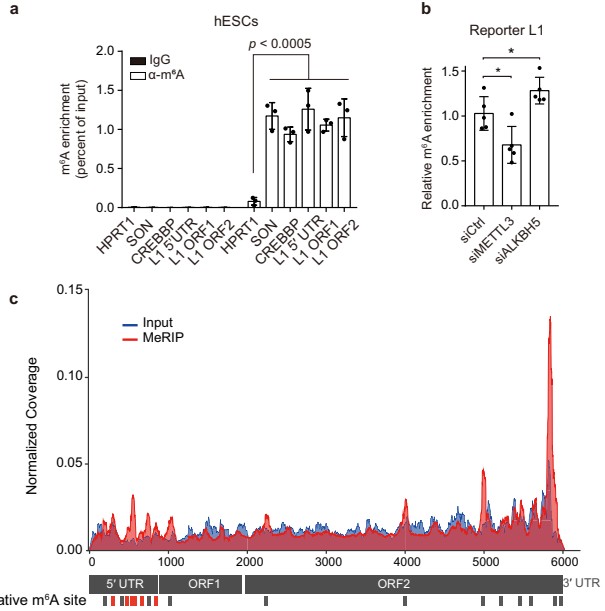

**Fig. 2 L1 RNA is modified by m⁶A. a** MeRIP-qPCR analysis of mRNA from H9 hESCs. Eluates from IgG immunoprecipitation served as negative control. Eluted RNA was quantified to determine the percentage of input. (*n* = 3 independent samples, mean ± s.e.m., one-way ANOVA and Dunnett's multiple comparisons test). **b** MeRIP-qPCR analysis of pL1Hs-transfected HeLa cells with m⁶A machinery knockdown. Eluted RNAs were quantified using primers specific for reporter L1. The enrichment of RNA was normalized to that of the control. (*n* = 5 independent samples, mean ± s.e.m., unpaired two-tailed *t*-test, \**p* < 0.05). **c** Map of m⁶A modification sites in full-length L1Hs from previously reported MeRIP-seq data for H1 hESCs (GSE52600). Read coverage was normalized to the total number of reads mapped to the L1Hs consensus sequence. The plot presents data from MeRIP-seq in red and input RNA-seq in blue. Bars (in red or black) indicate the m⁶A peaks identified by manual inspection in two replicates. m⁶A peaks in red correspond to peaks containing high score m⁶A-prediction sites. Source data are provided as a Source data file.

found in 3′ UTR and CDS, but not in the 5′ UTR (Fig. S3b). By mapping the reads on reporter L1, we obtained five candidate peaks (Fig. S3c), and further sorted according to the approach based on the m⁶A prediction as described above. All five peaks were classified as m⁶A-putative regions with high scores (Table S1b). Two of the featured peaks were located in the 5′ UTR, the other two were located in the ORF1, and another was in ORF2. The m⁶A modification sites commonly detected in endogenous and exogenous L1 RNA are A332 and A839, both located in the 5′ UTR (Table S1). This is a notable phenomenon since m⁶A modification typically occurs near the stop codon and at the 3′ UTR, and this gives rise to the possibility that the L1 5′ UTR acts as the regulatory hub for L1 mobility via m⁶A modification.

**5′ UTR m⁶A cluster is critical for L1 activity.** Given that the L1 5′ UTR has a potential m⁶A cluster, we next examined whether the L1 5′ UTR is necessary for m⁶A-dependent L1 regulation. We transfected the 5′ UTR-deleted pL1Hs (pL1Hs Δ5′ UTR) into HeLa cells treated with m⁶A machinery-targeting siRNA and monitored ORF1p expression. Intriguingly, the knockdown of m⁶A enzymes did not affect ORF1p expression in the absence of 5′ UTR (Fig. S4a). Furthermore, using the codon-optimized synthetic L1 construct that encoded ORFs with the same amino acids yet different nucleotide sequences, we

examined whether alterations in ORF1 and ORF2 nucleotide sequences could affect m⁶A machinery-mediated L1 regulation. Remarkably, silencing of METTL3 or ALKBH5 regulates L1 ORF1p expression only when 5′ UTR is contained in synthetic L1, which indicates that m⁶A machinery regulates L1 expression in a 5′ UTR-dependent manner (Fig. S4b). These results suggest that the L1 5′ UTR contains functional m⁶A motifs for successful ORF1p expression.

To identify the site of functional m⁶A in L1 5′ UTR, we selected six adenosine candidates of m⁶A modification (332, 495, 569, 600, 679, and 839, numbering based on L1PA1 consensus sequence[32]) through MeRIP-seq analysis in either hESCs or L1-reporter-expressing HeLa cells (Table S1). We generated a set of firefly luciferase reporter plasmids encoding L1 5′ UTR or its m⁶A-silencing A to T mutants (Fig. 3a). To quantify the effect of L1 5′ UTR m⁶A mutation without the bias from transfection efficiency, we normalized the firefly luciferase activity to that of *Renilla* luciferase. The dual-luciferase reporter assay revealed that a single A332T, A495T, or A600T mutation reduced the expression of firefly luciferase, compared to that of native 5′ UTR (Fig. 3b). However, the weak effect of these single mutants led us to hypothesize that multiple m⁶A modifications may function synergistically. Indeed, the double mutation of A332/600T and the triple mutation of A332/495/600T exerted significantly more synergistic and potent effects (Fig. 3c).

We next performed the L1 retrotransposition assay using the 5′ UTR m⁶A mutants of the pL1Hs construct. Mutations at each m⁶A motif of A332, A495, and A600 showed a marginal effect on L1 retrotransposition, whereas A332/A495/A600 triple mutation (hereinafter referred to as pL1 m⁶A mut) markedly inhibited L1 mobility (Fig. 3d and Fig. S5a–c). We validated the effect of the m⁶A cluster using the L1-luciferase reporter construct pYX014. Indeed, the triple m⁶A mutant of the L1-luciferase construct (pYX014 L1 m⁶A mut) induced approximately 50% decline in L1 mobility compared to that induced by the wild-type L1 (Fig. S5d).

To assess the effect of the triple mutation in the m⁶A modification level of L1, we performed MeRIP-qPCR for comparing m⁶A enrichments between cells that expressed pL1Hs and pL1 m⁶A mut. Surprisingly, the triple mutation reduced the enrichment of m⁶A-modified L1 by ~50%, while it did not affect the m⁶A levels of the endogenous controls SON and CREBBP (Fig. 3e and Fig. S5e). These results indicate that A332, A495, and A600 are the essential adenosines for L1 mobility and serve as m⁶A modification sites.

Based on our finding that ALKBH5 inhibits L1 mobility, we attempted to determine whether ALKBH5 could restrict the mobility of the L1 m⁶A mutant. L1 assays with co-transfection of pL1 vectors and FH-ALKBH5 revealed that the ectopic expression of ALKBH5 impaired the retrotransposition of pL1Hs (Fig. 3f, and Figs. S1e, 5f). However, ALKBH5 overexpression caused only marginal effects in pL1 m⁶A mut-expressing cells (Fig. 3f). Moreover, silencing the triple m⁶A modification led to the suppression of L1 mobility in AcGFP-expressing cells, but not in ALKBH5-expressing cells (Fig. 3f). In a reciprocal experiment, we measured the L1 retrotransposition frequency of pL1Hs and pL1 m⁶A mut in ALKBH5-lacking cells. Notably, ALKBH5 knockdown led to the enhancement of L1 mobility in pL1Hs-expressing cells, whereas no measurable changes were observed in pL1 m⁶A mut-expressing cells (Fig. 3g, and Fig. S5g, h). Consistent with this result, ALKBH5 was not able to suppress L1 ORF1p expression in the absence of the m⁶A cluster (Fig. S5i, j). In summary, we demonstrated that ALKBH5 suppresses L1 expression in the 5′ UTR m⁶A cluster-dependent manner, which suggests that the L1 5′ UTR m⁶As serve as the substrates for ALKBH5 demethylation.

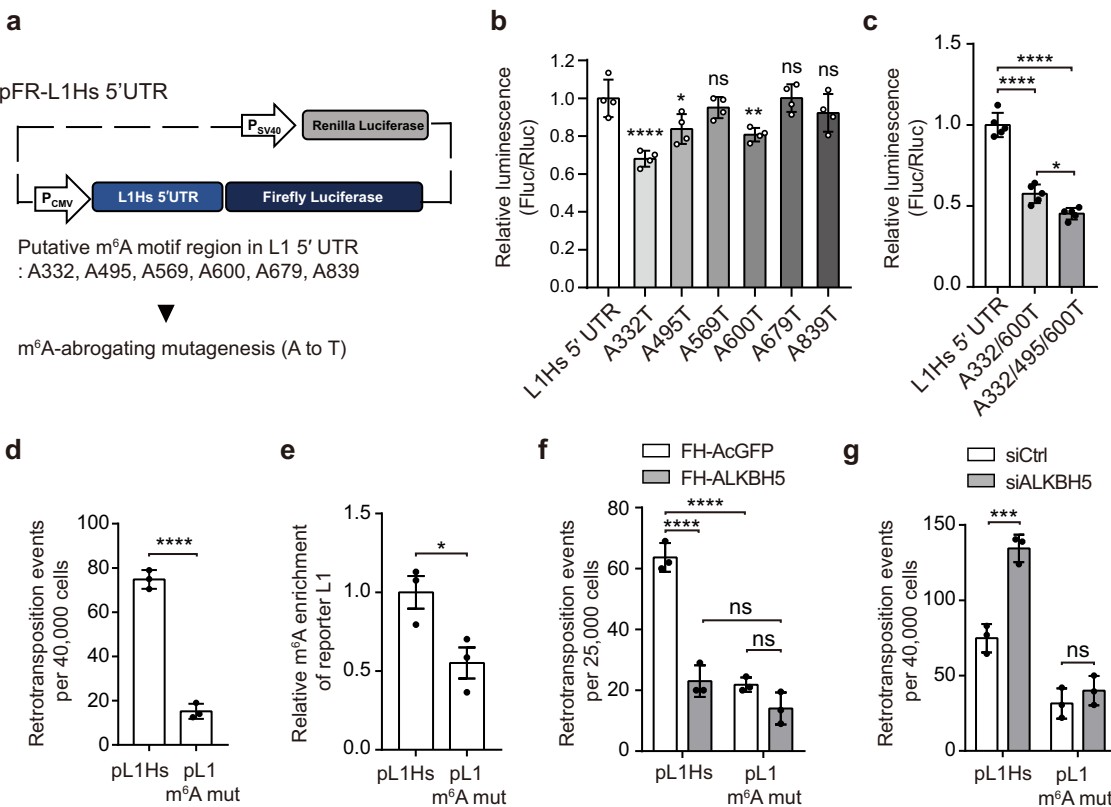

**Fig. 3 L1 5′ UTR m$^6$A cluster promotes L1 activity. a** Schematic of the dual-luciferase plasmid carrying L1 5′ UTR upstream of the firefly luciferase gene (pFR-L1Hs 5′ UTR). Firefly luciferase luminescence reflected the effect of 5′ UTR and of its mutations. **b**, **c** Dual-luciferase assay using HeLa cells transfected with pFR-L1Hs 5′ UTR or its A to T m$^6$A-abrogating mutant. The ratio of the luminescence of firefly and *Renilla* luciferase (Fluc/Rluc) was normalized to pFR-L1Hs 5′ UTR-expressing cells. (mean ± s.d., four (**b**) or five (**c**) independent samples). **d** L1 assays using the triple m$^6$A mutated L1 construct (pL1 m$^6$A mut) in HeLa cells ($n = 3$ independent samples, mean ± s.d.). **e** MeRIP-qPCR analysis for evaluating the effect of the triple m$^6$A mutation construct (pL1 m$^6$A mut). m$^6$A antibody-bound L1 RNA was normalized to that of pL1Hs-transfected cells. ($n = 3$ independent samples, mean ± s.e.m.) **f**, **g** Retrotransposition assay using pL1Hs-or pL1 m$^6$A mut-expressing HeLa cells ALKBH5 overexpression (**f**) or silencing (**g**) ($n = 3$ independent samples, mean ± s.d.). In **b**–**g**, Statistical significance was calculated by one-way ANOVA with Dunnett's (**b**), Tukey's multiple comparisons test (**c**, **f**, **g**), and unpaired two-tailed *t*-test (**d**, **e**) (****$p < 0.0001$, ***$p < 0.001$, and **$p < 0.01$, *$p < 0.05$, ns not significant). Source data are provided as a Source data file.

**m$^6$A modification promotes the translational efficiency of L1 RNA.** Given that m$^6$A regulates L1 ORF1p expression, we investigated the stages in the L1 replication cycle that are regulated by m$^6$A modification. First, we quantified L1 RNA expression in the presence or absence of the 5′ UTR m$^6$A cluster using two different plasmids, pL1Hs and pYX014. Irrespective of the vectors used, L1 m$^6$A mutation did not influence the levels of L1 RNA expression through northern blot and qRT-PCR (Fig. S6a–c). We next assessed the stability of reporter L1 mRNAs with or without the 5′ UTR m$^6$A mutation using the transcription inhibitor, actinomycin D. L1 RNA was more stable in both pL1Hs- and pL1 m$^6$A mut-expressing HeLa cells when compared to positive control, c*MYC* mRNA (Fig. S6d). We did not observe any significant difference in L1 RNA stability by m$^6$A mutation (Fig. S6d). We next examined the distribution of reporter L1 mRNAs in the nuclear and cytoplasmic fractions. In comparison to that of *GAPDH* (abundant in the cytoplasm) and *MALAT1* (abundant in the nucleus), over 80% of the L1 mRNA was present in the cytoplasmic fraction and the m$^6$A-deficient mutation did not affect the cellular localization of L1 RNA (Fig. S6e).

Several recent studies have linked 5′ UTR m$^6$A modification to translational efficiency in the context of cellular stress[34–36]. Besides, a previous study raised the possibility that the presence of the L1 5′ UTR determines the quality of L1 RNA[37]. Therefore, we

reasoned that the L1 5′ UTR m$^6$A cluster could modulate the translation of L1 RNA. To test this hypothesis, we performed an immunoblot assay in HeLa cells that expressed a single to triple m$^6$A mutant of the pL1 construct. The expression levels of ORF1p gradually decreased as the number of mutations increased (Fig. 4a and Fig. S7a). In addition, through polysome profiling, we captured polysome-bound RNA to assess the translational efficiency of L1 RNA. The deletion of the m$^6$A cluster significantly reduced the enrichment of polysome-bound L1 RNA compared to that of pL1Hs (Fig. 4b and Fig. S7b). To validate these results, we investigated whether m$^6$A regulates the translational efficiency of endogenous L1 mRNAs in PA-1 human embryonic carcinoma cells. Consistent with the effects of m$^6$A machinery depletion in pL1Hs-expressing HeLa cells (Fig. 1e), ALKBH5 knockdown augmented the production of endogenous ORF1p while METTL3 knockdown reduced ORF1p synthesis (Fig. 4c and Fig. S7c). The comparable levels of L1 mRNA in PA-1 cells with or without ALKBH5 depletion suggests that the enhanced production of ORF1p is a consequence of translational upregulation (Fig. S7d). Consistent with this result, the levels of polysome-associated L1 RNA substantially increased in ALKBH5-depleted PA-1 cells in comparison to the control cells (Fig. 4d and Fig. S7e), which indicates that ALKBH5 regulates L1 retrotransposition by suppressing the efficiency of L1 RNA translation.

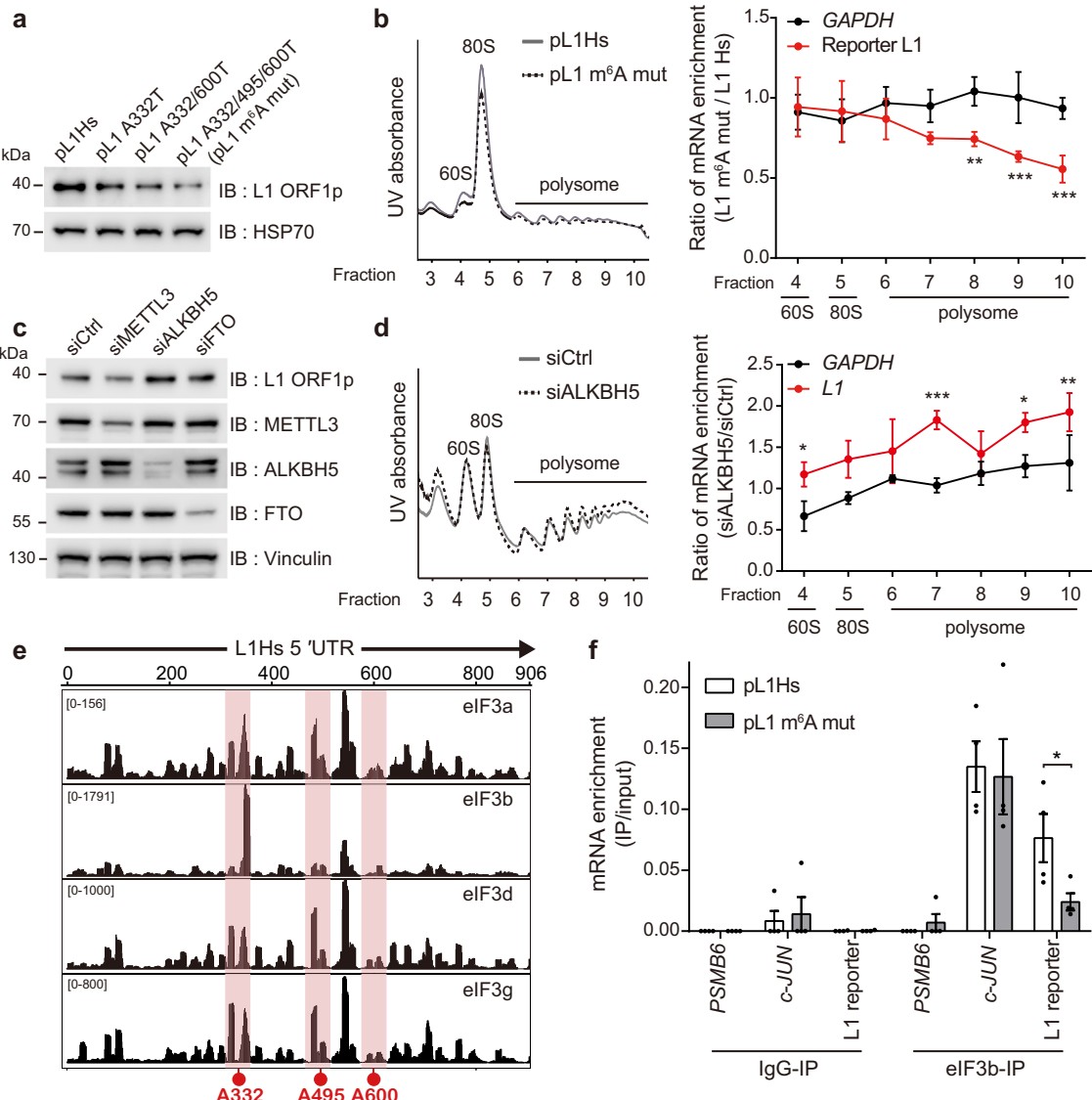

**Fig. 4 L1 5′ UTR m⁶A cluster enhances the translational efficiency through the recruitment of eIF3. a** Immunoblot analysis for assessing the effect of m6A mutation in L1 ORF1p levels. HSP70 served as a loading control. The immunoblot images are representative of three independent experiments. **b** Polysome profiling of pL1Hs- or pL1 m⁶A mut-expressing HeLa cells (left panel). Ratio of the polysome-bound mRNA levels in pL1 m⁶A mut-expressing cells to those in pL1Hs-expressing cells (right panel). The levels of RNA in each polysome fraction were normalized to the spike-in control and to the levels of input RNA. ($n = 4$ independent samples, mean ± s.d., two-way ANOVA and Bonferroni's multiple comparisons test; ***$p < 0.001$, **$p < 0.01$, and *$p < 0.05$ in comparison to the enrichment ratio of *GAPDH* in each fraction). **c** Immunoblot assay for determining endogenous L1 ORF1p levels in PA-1 cells treated with indicated siRNAs. Vinculin served as a loading control. Quantification of L1 ORF1p levels is shown as values normalized to those of Vinculin in Supplementary Fig. 7C. The immunoblot images are representative of four independent experiments. **d** Polysome profiling of PA-1 cells lacking ALKBH5 compared to siCtrl (left panels). The levels of endogenous L1 RNA was measured as in **b** using L1 5′ UTR-specific primers (right panel). ($n = 3$ independent samples, mean ± s.d., statistical significance was determined as in **b**). **e** Identification of eIF3-binding sites in L1Hs 5′ UTR using the previously reported eIF3 PAR-CLIP data set (GSE65004)[38]. The red boxes indicate the m⁶A sites-containing region. **f** eIF3 UV-CLIP-qPCR using pL1Hs- or pL1 m⁶A mut-expressing HeLa cells. IgG-IP and *PSMB6* served as negative controls ($n = 4$ independent samples, mean ± s.e.m., unpaired two-tailed *t*-test; *$p < 0.05$). Source data are provided as a Source data file.

eIF3 is an m⁶A-binding protein and promotes the selective translation of mRNAs that bear m⁶A in 5′ UTR[34]. These characteristics of eIF3 lead us to hypothesize that the L1 5′ UTR m⁶A cluster serves as a docking site for eIF3 to promote translation. To define the functional relationship between eIF3 and the L1 m⁶A cluster, we analyzed previously reported data from photoactivatable ribonucleoside-enhanced crosslinking and immunoprecipitation sequencing (PAR-CLIP seq) of eIF3 subunits a, b, d, and g[38]. By mapping the reads from PAR-CLIP of the eIF3 subunits along the endogenous L1Hs, we

revealed that eIF3 exhibits preferential binding to the L1 5′ UTR (Fig. S8a). Furthermore, the PAR-CLIP clusters were significantly enriched in the A332 m⁶A region in all four eIF3 subunits, while the A495 m⁶A region contained PAR-CLIP clusters of three eIF3 subunits: eIF3a, d, and g (Fig. 4e). We were unable to detect the comparable eIF3-binding sites in the A600 m⁶A region (Fig. 4e). To verify the interaction between eIF3 and the L1 m⁶A cluster, we transfected pL1Hs or pL1 m⁶A mut into HeLa cells and performed UV crosslinking immunoprecipitation using eIF3b antibody (Fig. S8b). Through RT-qPCR analysis of the

immunoprecipitated eluates, we observed the enrichment of L1 RNA comparable to *c-JUN*, a known eIF3-bound mRNA, in pL1Hs-expressing cells (Fig. 4f). *PSMB6* and eluates from IgG immunoprecipitation served as negative controls. Remarkably, the silencing of the m⁶A cluster reduced the quantity of eIF3-bound L1 RNA by ~70%, which indicates that the L1 5′ UTR m⁶A cluster bears the eIF3 docking site (Fig. 4f). Indeed, eIF3 knockdown suppressed endogenous L1 ORF1p expression in PA-1 cells and L1 retrotransposition in HeLa cells (Fig. S8c, d).

Since another m⁶A binding protein, YTHDF1, regulates translation efficiency of m⁶A-modified RNA and interacts with eIF3[39], we tested whether YTHDF1 also binds to L1 5′ UTR m⁶A cluster. Through RNA immunoprecipitation and qPCR analysis, we confirmed that L1 RNA associates with YTHDF1 and another YTH protein, YTHDF2, (Fig. S8e, f). However, 5′ UTR m⁶A cluster mutation did not impair interaction between YTHDFs and L1 RNA (Fig. S8f). These data suggest that YTHDFs bind to L1 RNA via m⁶A in region other than 5′ UTR. Collectively, the 5′ UTR m⁶A cluster specifically recruits eIF3 for the efficient translation of L1 RNA.

**5′ UTR m⁶A cluster is necessary to produce a functional unit for L1 retrotransposition.** For successful L1 retrotransposition, both ORF1p and ORF2p are required to generate the L1 RNP with the encoding L1 RNA[40]. Though we observed m⁶A-mediated regulation in ORF1p synthesis, it is necessary to determine whether m⁶A modification at the 5′ UTR influences L1 RNP formation. To investigate L1 RNP regulation by m⁶A, we obtained the cellular RNP fraction as previously reported[41]. Briefly, we prepared lysates from pL1-transfected cells and purified L1 RNPs using sucrose cushion ultracentrifugation (Fig. 5a). We detected comparable levels of L1 RNA in the RNP fractions from pL1Hs- and pL1 m⁶A mut-expressing cells (Fig. 5b and Fig. S9a). cDNA synthesis reaction in absence of reverse transcriptase revealed that neither genomic DNA nor plasmid contamination was present in the RNP fraction (Fig. 5b). Immunoblotting of the RNP fraction showed that the levels of RNP-associated ORF1p were diminished by L1 5′ UTR m⁶A mutation (Fig. 5c). This indicates that the m⁶A cluster mutation abolished the sufficient production of ORF1p for L1 RNP formation.

Since ORF2p expression level is too low to observe changes in the m⁶A mutant[42,43], we introduced the L1 element amplification protocol (LEAP) to gauge the reverse transcriptase activity of ORF2p[41] (Fig. 5a). Incubation of RNPs with LEAP primer facilitates ORF2p-mediated L1 cDNA synthesis. We amplified LEAP products using PCR with specific primers for reporter L1 and RACE adapter, which yielded products of 300–400 base pairs (bp) (Fig. 5d). However, m⁶A-abrogated L1 RNP produced cDNA at significantly lower levels than the wild-type L1 RNP did (Fig. 5d). These results reveal that the m⁶A cluster is necessary for L1 cDNA production, which suggests that the m⁶A cluster regulates ORF2p expression or its activity.

ORF1p oligomerization is critical for successful L1 retrotransposition[44]. We examined whether inefficient ORF1p synthesis results in a failure of L1 RNP formation. For a quantitative assessment of individual L1 RNP formation, we introduced the pAD3TE1 construct carrying T7-tagged ORF1p and MS2-stem-loop structures in the L1 3′ UTR (Fig. 5e). We performed RNA fluorescence in situ hybridization (FISH) with fluorescent Q670-labeled probes complementary to the linker regions between the MS2 loops and immunofluorescence experiments with anti-T7 antibody (Fig. 5f and Fig. S9b). Through z-stack analysis, we obtained the coordinates for the fluorescent signals of L1 RNA and ORF1p and identified the L1 RNPs by sorting out colocalizing particles within an intermolecular distance of 330 nm between L1 RNA and ORF1p. Consistent with the previous study[45], we observed colocalizing signals of L1 RNP as cytoplasmic aggregates (Fig. 5f). However, L1 m⁶A mut-expressing cells showed a significant reduction in both the number of L1 RNP foci and the signal intensity of colocalizing ORF1p (Fig. 5f-h). These data indicate that the abrogation of the m⁶A cluster reduces the levels of ORF1p in L1 RNP and causes a concomitant decrease in the number of L1 RNP particles.

**m⁶A is a driving force for L1 evolution.** Over the last 40 million years of human evolution, L1 subfamilies have frequently acquired novel 5′ UTRs[32]. Since a new L1 lineage will emerge only through its successful replication, the genetic novelty that promotes L1 mobility must remain preserved in the genomic fossils of L1s[46]. Considering that RNA methyltransferase installs m⁶A in a sequence-specific manner, we speculated that nucleotide mutations might lead to the acquisition or loss of the m⁶A consensus motif during L1 evolution. To unravel the evolutionary history of L1 5′ UTR m⁶A cluster regions, we analyzed 443 human-specific full-length L1s[47] and compared the three m⁶A motif sites, A332, A495, and A600. Given that adenosine residue should be followed by cytosine residue to form the m⁶A consensus motif DRA^mCH, A332 m⁶A-positive L1s constitute a considerably small part in the L1PA3 lineage (12.4%). In L1PA2 and younger lineages, the number of A332 m⁶A positive L1s increased drastically (92.9%) (Fig. 6a), and the same was observed in the youngest L1Hs (Fig. S10a). On the contrary, A495 and A600 are tightly conserved in all human-specific L1 subfamilies (Fig. S10b). We investigated this tendency of the A332 m⁶A motif in L1s of chimpanzee and gorilla, which share L1PA2 and L1PA3 lineages with humans. Comparative analysis of chimpanzee- or gorilla-specific full-length L1s revealed the seismic shift toward the population of A332 m⁶A positive L1s, while the chimpanzee- or gorilla-specific L1s continue to harbor the m⁶A motifs of A495 and A600 (Fig. 6a, and Fig. S10c-f). As in the L1Hs subfamily, the majority of the youngest chimpanzee-specific L1 subfamily (L1Pt) harbor the A332 m⁶A motif (Fig. S10C). In summary, we found that A332 m⁶A motif acquisition by single nucleotide substitution (T333C) first appeared in L1PA3 or older lineages, which indicates that the productive potential of m⁶A has allowed positive selection of A332 m⁶A-positive L1s during the evolution from the common ancestor.

To evaluate the consequence of A332 m⁶A acquisition in ancestral L1 5′ UTR, we generated a chimeric pL1 construct that contained L1PA2 5′ UTR and L1Hs ORF1/2 with the *mblastI* reporter (Fig. 6b). Based on the m⁶A consensus motif DRA^mCH, T333 of pL1PA2^5′UTR does not allow m⁶A modification at A332, whereas T333C point mutation enables A332 m⁶A modification (Fig. 6b). The retrotransposition assay revealed that T333C mutation enhanced the mobility of pL1PA2^5′ UTR, while the mutagenesis control (T333G) did not exert the same effect (Fig. 6c and Fig. S11a, b). As expected, the T333C m⁶A-gain mutation enhanced ORF1p synthesis of pL1PA2^5′ UTR (Fig. 6d). Although the acquisition of A332 m⁶A motif only led to a 1.4-fold increase in the cultured cell-based L1 retrotransposition assays (Fig. 6c), the 12 million years of L1 evolution would have been sufficient to amplify the profound effect of m⁶A. These results suggest that m⁶A modification in the L1 5′ UTR region may have played a crucial role in the L1 evolution of primates.

**Discussion**

The role of m⁶A modification in pathogenic viral transcripts has been reported in the past decade[48]. However, the role of m⁶A in L1s as genomic parasites have been poorly understood. In our

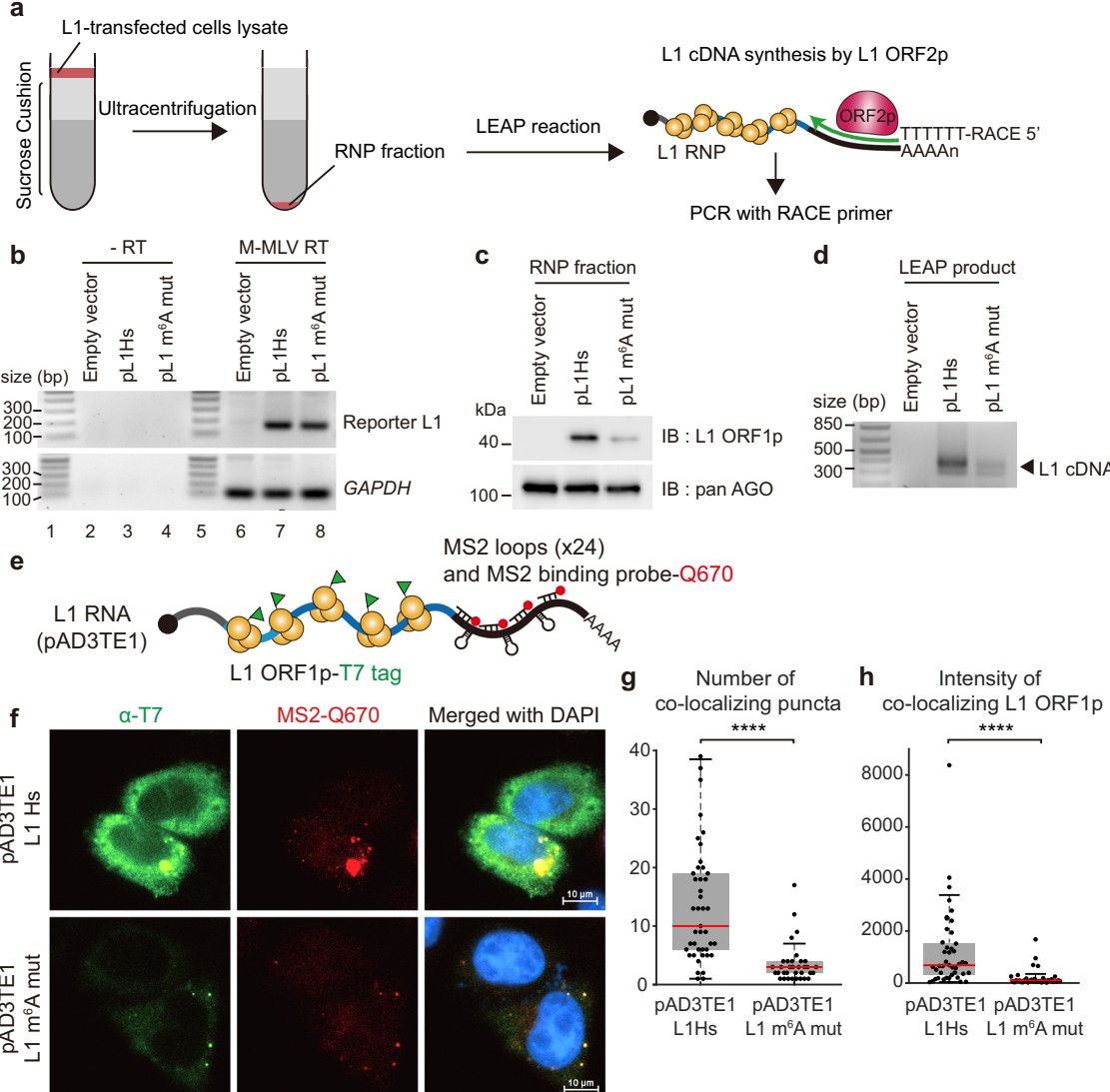

**Fig. 5 m6A modification is crucial for generating retrotransposition-competent L1 RNPs. a** Scheme of LEAP assay with description. Ultracentrifugation and purification cellular RNP (left part of scheme, cell lysate in red, and sucrose cushion in gray). LEAP reaction (right part of scheme, L1 ORF1p in yellow, L1 ORF2p in magenta, 3′ RACE adapter in green). **b** Quantification of mRNA levels in the RNP fraction of pL1-expressing HeLa cells. cDNA synthesis in the absence of reverse transcriptase (lane 2–4) and transfection of empty vector (lane 6) served as negative controls. The RT-PCR products of reporter L1 and *GAPDH* are of 158 and 106 bp, respectively (lane 6–8). Lane 1 and 5 show the DNA ladder. **c** Immunoblot assay of the RNP fraction from pL1-expressing HeLa cells. pan AGO served as a loading control. **d** LEAP assay using RNP fraction from pL1-expressing HeLa cells. The LEAP product is a diffuse band of 300–400 bp. The images of **b**–**d** are representative of three independent experiments. **e** A schematic of the L1-MS2 construct (pAD3TE1) carrying T7-tagged ORF1p (green) and MS2-stem-loops with Q670-labeled MS2 binding probes (red). **f** Immunofluorescence and RNA FISH images depicting HeLa cells transfected with pAD3TE1 L1Hs (top) or L1 m6A mut (bottom). Images for T7-tagged ORF1p (green), L1-MS2 RNA (red), and the merged images with DAPI (blue) are indicated. The images are representative of two independent experiments. **g** The number of L1 RNP foci in pAD3TE1-expressing HeLa cells. Colocalizing puncta within an intermolecular distance of 330 nm were counted as L1 RNP using z-stack analysis. **h** Intensity of L1 ORF1p in colocalizing puncta. Each point represents the intensity of L1 ORF1p per cell. **g**, **h** Boxplots indicate median (red middle line), 25th, 75th percentile (gray box) and 5th and 95th percentile (whiskers). 43 cells for L1Hs and 34 cells for L1 m6A mut, two-sided Kolmogorov–Smirnov test, ****$p < 0.0001$. Source data are provided as a Source data file.

study, we demonstrated that the proper formation of the m6A cluster in 5′ UTR of L1 RNA is essential for L1 retrotransposition. The evolutionary history of the m6A cluster in primate-specific L1s revealed the most influential m6A region (A332) that was obtained in the past 12 million years. This suggests the potential role of m6A as a driving force in L1 evolution (Fig. 7).

Two recent studies have revealed that the m6A modification decreases the stability of L1 RNA with respect to R-loop or chromatin regulation[27,28]. However, our results revealed that the L1 5′ UTR m6A cluster did not affect RNA stability but promoted

translation. Abakir et al., and Liu et al. observed the role of m6A in genome-wide L1 repetitive elements, which are mostly inactive by 5′ truncations or inversions[3,49]. Considering that our study focused on the functions of m6A in the replication cycle of retrotransposition-competent L1s, which have intact 5′ UTR, this difference in the scope of L1 RNA types may contribute to the discrepancy.

Indeed, m6A enzymes regulate L1 expression only when L1 contains its 5′ UTR. The presence of 5′ UTR in L1 transcripts affects retrotransposition efficiency[37]. Despite the unique

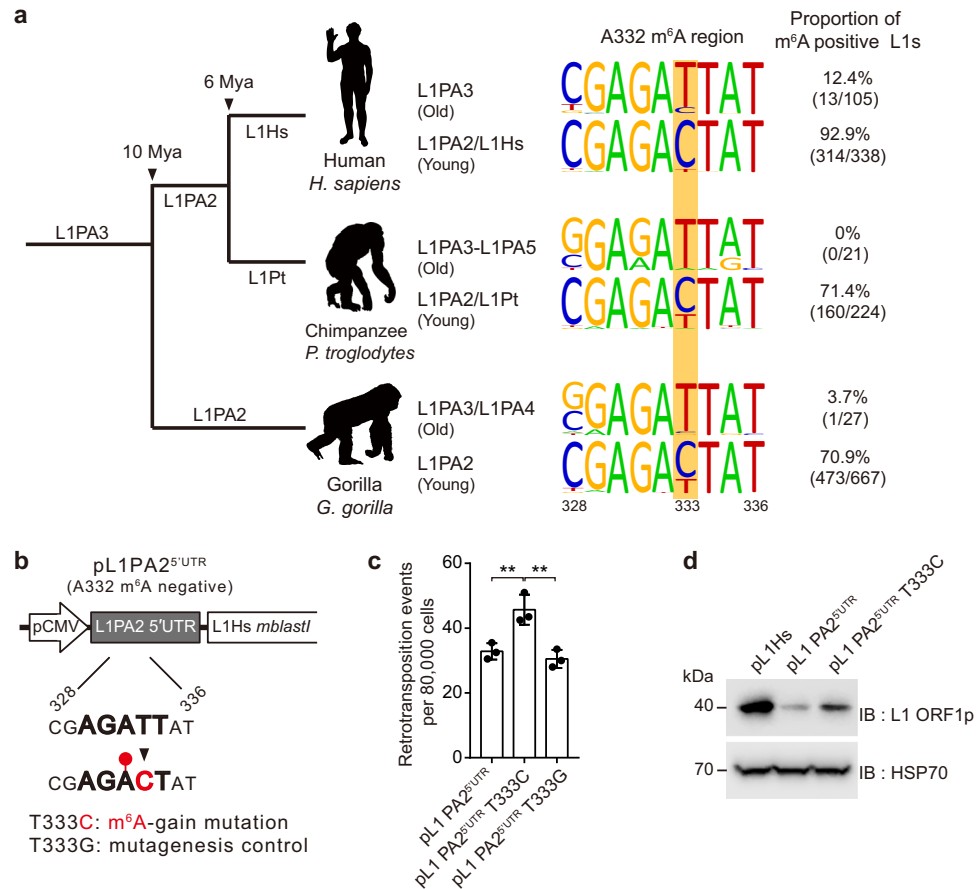

**Fig. 6 m6A is a driving force in L1 evolution. a** Comparative analysis of L1 A332 m6A sites in species-specific full-length L1s from three primates. Phylogenetic tree of gorilla, chimpanzee, and human L1s with predicted age and the corresponding L1 subfamily lineages (left). Changes in the A332-m6A motif region from L1PA3 or older L1s to L1PA2 and a younger L1 (right). The substitution site wherein the residue converts from T to C (333) is highlighted in yellow. The percentage indicates the proportion of m6A motif-positive L1s with nucleotide C to total L1s. **b** A schematic of retrotransposition assay using pL1PA2⁵′ UTR construct that is generated by substituting 5′ UTR of pL1Hs with A332 m6A negative 5′ UTR of L1PA2. A schematic of T333C m6A acquisition mutagenesis in L1 5′ UTR 328–336 region was indicated in red. **c** Retrotransposition assays for assessing the effect of A332 m6A acquisition in pL1PA2⁵′ UTR with T333C mutation. T333G mutation served as negative control ($n = 3$ independent samples, mean ± s.d., one-way ANOVA and Tukey's multiple comparison test; **$p < 0.01$). **d** Immunoblot assay showing L1 ORF1p expression in the indicated pL1-transfected HeLa cells. HSP70 served as a loading control. The immunoblot images are representative of three independent experiments. Source data are provided as a Source data file.

characteristics of L1 5′ UTR that is lengthy, GC rich, and exhibits promoter activity, its regulatory function at the post-transcriptional level has posed a long-standing question. Our findings demonstrated that L1 5′ UTR m6A modification is essential for L1 translation, L1 RNP formation, and thus retrotransposition. Therefore, we provide a new perspective on the regulatory function of L1 5′ UTR as a hub for RNA modification.

We demonstrated that m6A promotes not only ORF1p production via enhancing the translational efficiency, but also L1 cDNA synthesis. Since ORF2p can proceed reverse transcription regardless of association with ORF1p[41,45], it remains to clarify whether m6A modification upregulates ORF2p translation or m6A- modified L1 RNA indirectly influences reverse transcriptase activity of ORF2p. The unconventional translational mechanism of ORF2p, which relies on the translation of the upstream ORF[50], suggests that enhanced ORF1p translation rates by m6A cluster successively stimulate ORF2p synthesis. In addition, m6A modifications could alter RNA-protein interactome[51,52] or RNA secondary structure[53], which might affect L1 ORF2p enzymatic activity. Therefore, future studies could reveal the role of m6A in ORF2p regulation. By adopting a microscopic approach, we confirmed that m6A is critical for the formation of L1 RNP aggregates. The rate of ORF1p oligomerization is the limiting

factor in the production of successful L1 RNPs[44]. Therefore, we speculated that 5′ UTR m6As enable L1 RNA to produce sufficient ORF1p, which further accelerates the oligomerization of ORF1p. Since the process of L1 RNP formation is more complicated than the biochemical interaction between L1 RNA and its protein, the process by which m6A orchestrates the assembly of retrotransposition-competent L1 RNP remains to be understood.

eIF3 recognizes an m6A residue in the 5′ UTR and promotes the translation of mRNAs[34]. We assumed that the L1 5′ UTR recruits eIF3 to the m6A cluster for efficiently translating the L1 mRNA. Indeed, eIF3 PAR-CLIP-seq data reveal the interaction between eIF3 and L1 5′ UTR m6A residue. We also demonstrated that the eIF3-bound portion of L1 decreases in the absence of the 5′ UTR m6A. A single m6A residue is sufficient to induce eIF3-mediated translation[34]. This could explain the synergetic effects of triple m6A residues in L1 5′ UTR, which suggests that each m6A residue can serve as a docking site for eIF3. Moreover, under cellular stress, the 5′ UTR m6A facilitates the cap-independent translation of mRNA[34–36]. These studies raise the possibility that m6A initiates the cap-independent translation of L1 RNA. Although Dmitriev et al. revealed that human L1 mRNA is translated in a cap-dependent manner, m6A modification was not considered in their experiments[54]. Therefore, in future studies, it

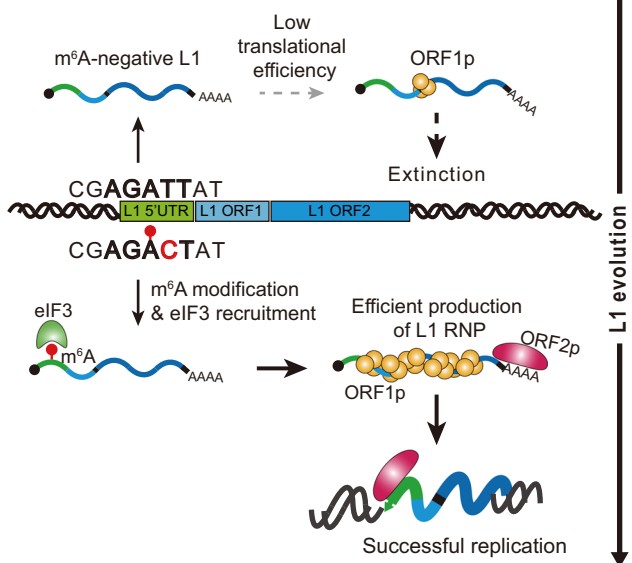

**Fig. 7 A proposed model for the role of m⁶A in L1 replication and evolution.** Full-length young L1s RNA have m⁶A in 5′ UTR A332 residue (lower part of scheme, m⁶A in red circle and m⁶A-gain mutation at T333C in red). 5′ UTR m⁶A cluster recruits eIF3 complex (green) and promotes translation of L1 ORF1p (yellow). Increased L1 ORF1p synthesis leads efficient production of L1 ribonucleotide particle (RNP) formation with its parental mRNA (line with poly A) and the reverse transcriptase, L1 ORF2p (magenta). The L1 RNP enters the nucleus and then generate the progeny through insertion of its cDNA. Old L1s with no A332 m⁶A motif have lower efficiency of translation and replication than those of A332 m⁶A-positive L1s (upper part of scheme). Since the A332 m⁶A motif first appeared ~12 million years ago, m⁶A-stimulated L1 replication has allowed m⁶A-positive L1s to survive during evolution, but made old L1s out of competition.

is important to determine whether m⁶A modification enables the cap-independent translation of L1 and whether m⁶A acts as a molecular switch for L1 expression under cellular stress.

L1s have been continuously active since the origin of mammals[55]. One of the previous studies on L1 evolution revealed that several distinct L1 lineages coexisted and were in a simultaneously activated state in the ancestral primate genome. However, since the emergence of the L1PA lineage, the L1 subfamily has evolved and maintained itself as a single lineage in the last 25 million years of the evolution of human and its close relatives[32]. The study proposed that the competition between or coexistence of L1 lineages is determined by the status of the 5′ UTR of L1s and acquisition of novel 5′ UTR is a fundamental feature in mammalian L1 evolution[32]. Given that m⁶A methyltransferase marks m⁶A in a sequence-specific manner, the accumulation of mutations in L1 might cause the loss or acquisition of putative m⁶A motifs. To further elucidate the history of m⁶A in L1 evolution, we analyzed species-specific full-length L1s from the human, chimpanzee, and gorilla genome. Notably, the A332 m⁶A motif first appeared in L1PA3 or older L1 lineages more than 12 million years ago. During the evolution of the three different primates, humans, chimpanzees, and gorillas, the A332 m⁶A-positive L1s have propagated their progenies and have become the dominant L1 subfamilies. As m⁶A modification promotes L1 mobility, the acquisition of m⁶A would have resulted in the positive selection of A332 m⁶A-containing L1s. Over the extended periods of L1 evolution, L1s have competed for survival against host restriction[15,16,32]. KRAB-zinc finger proteins (ZFP), which have evolved with L1s, suppress the old L1 transcription in a sequence-specific manner[15,16]. However, L1Hs, which is the youngest L1

lineage in the human genome, escapes KRAB-ZFP restriction and is not recognized by any KRAB-ZFPs[16]. Instead, the host defense utilizes post-transcriptional suppression mechanisms, such as small RNA interference (e.g., piRNA) or APOBECs, to restrict the replication of L1s[56]; however, the youngest L1s are still active. Our findings provide clues on how the youngest L1s continuously replicate under host surveillance. The emergence and the propagation of the A332 m⁶A-positive L1s suggest that 5′ UTR m⁶A modification was a countermeasure against the host post-transcriptional restriction.

## Methods

**Cells.** HeLa cells were cultured in DMEM supplemented with 10% (v/v) fetal bovine serum (FBS, HyClone) and 1% (v/v) GlutaMAXI (Gibco). Human embryonic carcinoma PA-1 cells were cultured in RPMI 1640 supplemented with 10% (v/v) FBS (HyClone) and 1% (v/v) GlutaMAXI (Gibco). hESCs (H9, Wicell Research) were cultured in defined hESC culture medium (Stem Cell Technology) on hESC-qualified extracellular matrix (Corning)-coated culture dishes (Corning) or on tissue culture wall plate (Falcon). The cultures were incubated at 37 °C in 5% CO₂.

**Plasmids.** The FLAG-HA-pcDNA3.1-derived plasmids used in this study were named using FH as a prefix with the respective protein names specified. AcGFP, ALKBH5, and FTO cDNA were cloned into FLAG-HA-pcDNA3.1 (Addgene, 52535) for overexpressing N-terminally FLAG-HA-tagged protein. FH-based plasmids were generated by restriction enzyme cloning using *XbaI* and *PmeI* (NEB). The site-directed mutagenesis of FH-ALKBH5 to catalytically inactive mutant (H204A) construct was performed using the Phusion High-Fidelity polymerase kit (Thermo Fisher Scientific).

pJJ101-L1-dn6 2.2, which is referred to as pL1Hs in this study, is a pCEP4-based plasmid that contains an active human L1(L1-dn6) and was generously provided by J. L. Garica-Perez[25]. For the mutagenesis of L1 5′ UTR m⁶A sites, the L1-dn6 5′ UTR and ORF1 region containing *NotI* and *AgeI* restriction sites was recloned into plasmid pCMV14. Using site-directed mutagenesis PCR, the following mutants of pCMV14 L1 5′ UTR ORF1 plasmids were prepared: Δ5′ UTR, A332T, A495T, A600T, A332/600T, and A332/495/600T (named as m⁶A mut). Next, *NotI-AgeI* fragments of pCMV14 L1 5′ UTR ORF1 mutant constructs were amplified and subcloned into pL1Hs. To generate pL1PA2⁵′ UTR, we synthesized the L1PA2 5′ UTR region based on reported consensus sequences using gene synthesis (Cosmogenetech). We then replaced the L1Hs 5′ UTR of pL1Hs with L1PA2 5′ UTR, as described above.

pAD3TE1 is an L1.3 plasmid containing the T7 gene 10 epitope tag on the carboxyl-terminus of ORF1p, TAP tag on the carboxyl-terminus of ORF2p, and 24 copies of the MS2 loop repeat in the 3′ UTR[45]. pAD3TE1 was gift from Aurélien J. Doucet. The generation of L1 5′ UTR m⁶A mutant constructs of pAD3TE1 was performed according to the method for pL1Hs mutant construct generation.

L1-firefly luciferase-tagged plasmids pYX014 and pYX015 were gifts from W. An[26]. pYX014 encodes L1 constructs under the L1 native 5′ UTR promoter. pYX015 carries a retrotransposition-defective mutation in L1 ORF1. pYX014 and pYX015 plasmids contain a *Renilla* luciferase cassette to normalize transfection efficiency levels. To generate pYX014 L1Hs and m⁶A mut constructs, *NotI-PmlI* fragments that were 2166 bp in length, including those spanning from the L1 5′ UTR to the forepart of ORF2 in pL1Hs and pL1 m⁶A mut, were subcloned into pYX014 via restriction enzyme cloning.

L1-neo-TET, a codon-optimized synthetic L1 construct, was generously provided by Astrid Roy-Engel (Addgene, 51284). The L1-neo-TET lacks a 5′ UTR. To generate a 5′ UTR-containing L1-neo-TET construct, the 5′ UTR of pL1Hs was amplified using PCR and the amplicon was inserted downstream of the CMV promoter of L1-neo-TET.

pFR-L1Hs 5′ UTR plasmids were generated by restriction enzyme cloning. The L1 5′ UTR of pL1Hs and firefly luciferase of pGL3-Basic (Promega) were cloned into pCMV14 downstream of the CMV promoter. Thereafter, the neomycin-resistant gene located downstream of the SV40 promoter was substituted with *Renilla* luciferase gene encoded by pYX014. Site-directed mutagenesis was used to generate the following m⁶A motif-abrogating mutants of pFR-based plasmids: A332T, A495T, A569T, A600T, A679T, A758T, A839T, A332/600T, and A332/495/600T.

pDEST HA-derived plasmids were named using HA as a prefix with the respective proteins, YTHDF1 and YTHDF2. YTHDF1 and YTHDF2 cDNA were cloned into pDEST HA vector using pENTR/D-TOPO vector (Invitrogen) and Gateway® LR Clonase® II Enzyme mix (Invitrogen). HA tag sequence is located in 5′ end of insert for overexpressing N-terminally HA-tagged protein.

**RNA interference.** siRNAs directed against METTL3 (L-005170-02), ALKBH5 (L-004281-01), FTO (L-004159-01), or nontargeting siRNAs (D-001210-01-50) were purchased from Dharmacon. All siRNA transfections were performed using the

DharmaFECCT 1 transfection reagent (Dharmacon) according to the manufacturer's instructions.

**Immunoblotting.** The cells were lysed in RIPA buffer (50 mM Tris pH 7.5, 150 mM NaCl, 1% Nonidet P-40 (NP-40), 0.5% sodium deoxycholate, 0.05% SDS, 1 mM EDTA, 1 mM DTT) supplemented with cOmplete protease inhibitor cocktail (Roche) for 15 min on ice. The lysates were cleared by centrifugation and mixed with Laemmli sample buffer. The mixture was then boiled at 98 °C for 10 min, separated by SDS-PAGE in 10% gels, and transferred onto nitrocellulose blotting membranes (Amersham). The membranes were blocked by incubating with 5% skim milk in Tris-Buffered Saline Tween-20 (TBST) for 30 min and incubated overnight at 4 °C with the respective primary antibodies at 1:1,000 dilution, except for anti-eIF3b antibody at 1:2000. Subsequently, the membranes were washed thrice with TBST and incubated with HRP-conjugated secondary antibodies (Peroxidase AffiniPure Goat Anti-Mouse IgG, 115–035–062, Jackson ImmunoResearch Laboratories or Peroxidase AffiniPure Goat Anti-Rabbit IgG, 111–035–003, Jackson ImmunoResearch Laboratories) at 1:5000 dilution in 5% skim milk/TBST. After washing thrice with TBST, the immunocomplexes were imaged using SuperSignal West Pico PLUS Chemiluminescent Substrate (Thermo Fisher Scientific). Band intensity quantification in Supplementary Fig. 7c were performed using ImageJ[57]. Uncropped blots were provided in the source data file.

**RNA extraction and RT-qPCR.** Total RNA was extracted using TRIzol reagent (Invitrogen) according to the manufacturer's instructions. For the removal of genomic or plasmid DNA, total RNA was treated with recombinant DNase I (Takara) for 1 h at 37 °C, followed by purification using the NucleoSpin RNA Clean-up kit (Macherey–Nagel). cDNA synthesis was performed using the ReverTra Ace qPCR RT Kit (Toyobo) according to the recommended protocol. TOPreal™ qPCR 2X PreMIX (Enzynomics) was used for subsequent qPCR reactions. The qPCR primers used in this study are listed in Supplementary Table 2.

**Polysome fractionation.** Ten milliliter of 10–50% linear sucrose gradients in base solution (100 mM NaCl, 20 mM Tris pH 7.5, 10 mM $MgCl_2$, 100 µg/ml of cycloheximide) were prepared a day before polysome fractionation. For polysome fixation, $7–10 \times 10^6$ cells were incubated for 10 min in a media containing 100 µg/ml of cycloheximide at 37 °C and were collected by scrapping with PBS containing 100 µg/ml of cycloheximide. After centrifugation at $1200 \times g$, 4 °C for 5 min, the cell pellets were lysed in 100 µl of polysome extraction buffer (20 mM Tris pH 7.5, 100 mM KCl, 5 mM $MgCl_2$, 0.5% NP-40) supplemented with RNase inhibitor (Enzynomics), protease/phosphatase inhibitor cocktail (Cell signaling), and 1 mM DTT. The cells were incubated in the buffer for 10 min at 4 °C and centrifuged at $12,000 \times g$, 4 °C for 10 min to remove debris and nuclei. The protein concentration was measured using the Pierce BCA Protein Assay kit (Thermo Fisher Scientific). Five hundred to 600 µg of the lysate was introduced at the top of the linear sucrose gradient and centrifuged at $222,000 \times g$, 4 °C for 2 h using the SW41Ti rotor of the Beckman ultracentrifuge. Fifty microgram of the lysate was saved as input RNA. After centrifugation, 1 ml fractions were collected from the top to the bottom of the gradient using the BioLogic LP system and fraction collector (BioRad) with UV absorbance at 260 nm. Next, 250 µl of each fraction was mixed with 750 µl of TRIzol LS reagent (Invitrogen) and 20 ng of spike-in RNA (synthesized firefly luciferase mRNA). RNA extraction and qPCR were performed as described above. The levels of RNA in each fraction were normalized to those of spike-in RNA and input RNA.

**L1 *mblastI* retrotransposition assay.** HeLa cells were plated at $8 \times 10^4$ cells in 12-well plates. After 18 h, the cells were transfected with L1 plasmid (pJJ101-L1-dn6 2.2; pL1Hs) at 800 ng per well using Lipofectamine 3000 (Invitrogen) following manufacturer's instructions. Two days later, 200 µg/ml hygromycin B (Invitrogen) was added to the media to select the transfected cells. Cell selection continued for 4 days, and the hygromycin B-resistant cells were reseeded at $2.5 \times 10^4$ per well in a 6-well plate. The next day, blasticidin S (Invitrogen) was added to a final concentration of 8 µg/ml and the cells were cultured for 7–9 days in its presence. The colonies were stained with crystal violet and counted using Colony, version 1.1 (Fujifilm). Retrotransposition assays were performed using RNA interference targeted toward METTL3, ALKBH5, and FTO with slight modifications in the process described above. For this, $6 \times 10^4$ HeLa cells were seeded into 12-well plates with 40 nM siRNA-Dharmafect1 (Dharmacon) mixture. After 24 h, the cultures were divided equally and plated into two wells in 12-well plates. The next day, the cells were transfected with pL1Hs at 500 ng per well. Four days after transfection, the cells were plated at $6 \times 10^4$ cells per well in 6-well plates and selected using 8 µg/ml of blasticidin S. Retrotransposition assays with the overexpression of AcGFP, ALKBH5, and FTO were performed as described above. Briefly, HeLa cells were transfected 500 ng FH-plasmid and 700 ng of L1 plasmid, and 4 days after transfection, $6 \times 10^4$ cells were reseeded into a well in a 6-well plate and selected after treatment with blasticidin S for 7–9 days.

**Luciferase assay.** HeLa cells were plated at $8 \times 10^4$ cells per well in 12-well plates. The next day, the cells were transfected with 800 ng per well of the pFR vector (pCMV-L1 5′ UTR-firefly luciferase) using Lipofectamine 3000 (Invitrogen). Two days later, the transfected cells were harvested and luminescence was measured using the Dual-Luciferase Reporter Assay System (Promega) according to manufacturer's instruction. Briefly, 250 µl of passive lysis buffer was used to lyse cells in each well in 12-well plates. Next, 20 µl of the lysate was mixed with 100 µl of the Luciferase Assay Reagent II, and the luminescence of firefly luciferase was measured using a microplate luminometer (BERTHOLD). *Renilla* luciferase activity was subsequently measured after administering 100 µl Stop & Glo Reagent.

**Crosslinking immunoprecipitation and qPCR (CLIP-qPCR).** eIF3-RNA CLIP-qPCR was performed as described previously[34] with some modifications. For each experiment, $1.2 \times 10^6$ HeLa cells were plated on two 100 mm dishes each. The next day, the cells were transfected with 6 µg of L1 plasmid per dish using Lipofectamine 3000 (Invitrogen). Two days later, the cells were washed twice with cold PBS, and allowed to form UV crosslinks on ice under 150 kJ/cm² of UV 254 nm light (XL-1500, Spectrolinker). The cells were scraped and transferred to PBS and pelleted by centrifugation at $1000 \times g$, 4 °C for 3 min. The pellets were resuspended in 1 ml of lysis buffer (50 mM Tris pH 7.5, 100 mM NaCl, 1% NP-40, 0.1% SDS, 0.5% sodium deoxycholate, 1× cOmplete protease inhibitor cocktail, 1 mM DTT, 80 unit/ml RNase inhibitor). The lysate was passed through a 21 G needle ten times and shock-frozen using liquid nitrogen. The lysate was thawed on ice and centrifuged at $15,000 \times g$ for 15 min. The supernatant was further cleared by filtering through a 0.22-µm membrane. From each lysate, 5% was retained as input. For immunoprecipitation, 10 µl of Dynabeads Protein G (Invitrogen) was washed twice with lysis buffer and incubated with 3 µg of eIF3b antibody (A301-761A, Bethyl) on a rotating wheel at room temperature for 1 h. The cell lysates were mixed with the antibody-bead complex and rotated overnight at 4 °C. The beads were washed five times in high-salt buffer (50 mM Tris pH 7.5, 500 mM NaCl, 1% NP-40, 0.1% SDS, 0.5% sodium deoxycholate, 1 mM EDTA, 1 mM DTT, 80 unit/ml RNase inhibitor). The antibody-lysate mixture and the conserved input lysates were resuspended in 100 µl of 1× Proteinase K buffer (100 mM Tris pH 7.5, 50 mM NaCl, 10 mM EDTA, 1% SDS). Next, 1 mg Proteinase K (Macherey–Nagel) was added into the suspensions. Protein digestion was conducted at 50 °C for 2 h in a shaking incubator. After incubation, 100 µl of 7 M Urea (w/v)-1X Proteinase K buffer was added into the immunoprecipitation samples, and the samples were re-incubated at 50 °C for 2 h in a shaking incubator. RNA was extracted using TRIzol LS supplemented with 20 ng of spike-in RNA.

**Methyl-RNA immunoprecipitation (MeRIP)-seq.** MeRIP was performed as described earlier[58] with some modifications. HeLa cells were plated on two 100 mm dishes at $1.2 \times 10^6$ cells per dish. After 18 h, the cells were transfected with 8 µg of pL1Hs per dish. After 48 h, poly (A) + RNA was extracted using the Poly (A) purist Mag kit (Invitrogen). The poly (A) + RNA was mixed with RNA fragmentation reagents (Invitrogen) and fragmented into oligonucleotide that was 50–150 nt in length by heating to 75 °C for 5 min. Fragmented RNA was purified by ethanol precipitation. Next, 6 µg of fragmented RNA was incubated with 4 µg of anti-m6A antibody (Merck, ABE572) in MeRIP buffer (50 mM Tris pH 7.5, 150 mM NaCl, 1 mM EDTA, and 0.1% NP-40) on a rotating wheel for 2 h at 4 °C. After that, the immunoprecipitation mixtures were mixed with Dynabead protein A (Invitrogen) and incubated overnight on a rotating wheel at 4 °C. After washing five times with MeRIP buffer, RNA was eluted twice by incubating in elution buffer on a rotating wheel for 1 h at 4 °C (6.7 mM m6A sodium salt and 200 unit/ml RNase inhibitor-containing MeRIP buffer). The eluted RNA was purified by ethanol precipitation. cDNA libraries were prepared as previously described[59]. Briefly, RNA was dephosphorylated using calf intestinal alkaline phosphatase (NEB) and labeled with γ-³²P-ATP using T4 polynucleotide kinase (Takara). RNA was separated by 10% urea-PAGE and purified from the excised gel corresponding to 50–150 nt RNA fragments. The extracted RNA was ligated to a 3′ adapter using T4 RNA ligase 2, truncated KQ (NEB). The RNA was then purified from free 3′ adapters by repeated gel excision. The 3′ adapter-ligated RNA was ligated to a 5′ adapter using T4 RNA ligase 1 (NEB) and subsequently reverse transcribed using SuperScript III reverse transcriptase (Invitrogen). The cDNA library was amplified by PCR using Phusion HF polymerase (Thermo Fisher Scientific), separated by 6% acrylamide gel electrophoresis, and purified by gel excision. The libraries were sequenced to $2 \times 100$ base-pair reads on the Illumina HiSeq 2500. The sequence of the 3′ and 5′ adapters, reverse transcription primer, and 5′ and 3′ PCR primers are listed in Supplementary Table 3. For MeRIP-qPCR analyses, we followed this procedure, except the poly (A) + RNA fragmentation step. Eluted RNA was purified using the NucleoSpin RNA Clean-up kit (Macherey–Nagel), and subjected to cDNA synthesis.

For MeRIP-seq analysis, the adapters were trimmed using Cutadapt[60] (cutadapt -g TACAGTCCGACGATC -A TGGAATTCTCGGGTGCCAAGG). The 3′ and 5′ adapter sequences in the first and second read in a pair (owing to the short insert size) were further trimmed and the read pairs with either reads < 18 bp were discarded. The remaining reads were then aligned to the combined human genome (hg19), and reporter L1 (pL1Hs) sequence using Spliced Transcripts Alignment to a Reference (STAR)[61] and peak calling was performed using MACS2[62]. For analyzing the m6A modifications in endogenous L1, the sequence reads from human embryonic stem cells were retrieved[31] (accession code: GSE52600) and

were aligned against L1Hs consensus sequence using STAR. The codes are available from https://github.com/hastj7373/merip-seq.

**RNA FISH and immunofluorescence.** The L1-MS2-stem-loop constructs pAD3TE1 L1Hs and pAD3TE1 L1 m⁶A mut were transfected into HeLa cells. The following day, the cells were reseeded on sterile coverslips where 200 μg/ml hygromycin B was added for selection. After 3 days, the cells were fixed with 3.7% formaldehyde in PBS for 10 min, and the fixation was quenched by adding 0.1 M glycine in PBS for 10 min. The fixed cells were permeabilized in 70% ethanol for at least 3 h to 1 week at 4 °C. The cells were then rehydrated with PBS for 30 min and incubated in a prehybridization solution (10% formamide, 2× SSC solution) for 30 min at 37 °C. Hybridization was performed overnight at 37 °C in 50 μL of hybridization solution containing 10% formamide, 2× SSC, 10% dextran sulfate, 50 μg yeast tRNA, 0.2% BSA, 0.1 M DTT, 50 units RNase inhibitor (Enzynomics), and 10 ng of MS2-Q670 probe (generously provided by Hye Yoon Park[63]; listed below) at 37 °C. Next, the cells were washed twice with a prehybridization solution for 30 min. For the immunofluorescence experiment, the hybridized cells were incubated in the blocking solution (10% formamide, 2× SSC, 0.2% BSA) for 1 h, followed by incubation with anti-T7 primary antibody (ab9138, Abcam) diluted in the blocking solution (1:200) for 2 h. The cells were washed twice with the prehybridization solution for 15 min and incubated with FITC-conjugated anti-goat secondary antibody (305–095–047, Jackson immunoresearch) diluted in blocking solution (1:200) for 1 h. The cells were washed twice as described above, and the coverslips were mounted on slide glasses using the Vectashield antifade medium with DAPI (Vector Laboratories). The samples were imaged using an inverted microscope Nikon Eclipse Ti2 equipped with a 1.45 numerical aperture Plan apo λ ×100 oil objective and a sCMOS camera (Photometrics prime 95B 25 mm). For each field of view, stacks of images of 6 μm were captured at every 0.3 μm in the DAPI395, GFP488, and Alexa647 channels using the NIS-Elements software.

The sequence of the RNA FISH probes are: MS2LK20 (5′ TTTCTAGAGTCG ACCTGCAG 3′), MS2 LK51-1 (5′ CTAGGCAATTAGGTACCTTAG 3′), and MS2 LK51-2 (5′ CTAATGAACCCGGGAATACTG 3′). Each probe was labeled with two Quasar 670 dyes at both ends. The mixture of the three probes were used for RNA FISH of L1 RNA tagged with the MS2 loops.

**Co-localization analysis of RNA FISH and IFA microscope image.** Binary masks of cells were generated using the ROI manager in ImageJ[57]. Protein and mRNA particles from z-stack images were detected using the TrackNTrace software[64]. After the detection of particles, the protein-mRNA pairs with an intermolecular distance of 330 nm (three pixels) were considered as colocalizing pairs. The intensities of proteins colocalizing with mRNA were determined based on the amplitude of the fitted 2D Gaussian function from the TrackNTrace software.

**LEAP assay.** The LEAP assay was performed as described previously[41]. Briefly, HeLa cells were plated ($4 \times 10^6$ cells in 60 mm dishes); the following day, the cells were transfected with 3 μg of L1 plasmid (pL1Hs) using Lipofectamine 3000 (Invitrogen). After 48 h, 200 μg/ml hygromycin B was added to the media to select the cells carrying the L1 plasmid. After 2 days of selection, the cells were lysed with CHAPS lysis buffer (10 mM Tris pH 7.5, 0.5% CHAPS (w/v), 1 mM MgCl₂, 1 mM EGTA, and 10% glycerol) supplemented with 1 mM DTT and the cOmplete protease inhibitor cocktail (Roche) and cleared by centrifugation (4 °C, $20,000 \times g$ for 15 min). The cleared lysates were loaded on a sucrose cushion (20 mM Tris pH 7.5, 80 mM NaCl, 8 mM MgCl₂, 1 mM DTT, 1× protease inhibitor, 4 ml of 8.5% sucrose (from the top) and 6 ml of 17% sucrose (from the bottom) solutions) in 13.2 ml Ultra-Clear tubes (Beckman Coulter) and centrifuged at $178,000 \times g$, 4 °C for 2 h in a SW41Ti rotor of Beckman ultracentrifuge. The colorless pellets were suspended by pipetting in 100 μL of RNase-free water supplemented with 1× protease inhibitors. Pierce BCA Protein Assay (Thermo fisher Scientific) was conducted to determine the protein concentration. Three microgram of the RNP samples were retained and used later in RNA isolation and immunoblotting experiments. Seven hundred and fifty nanogram of each RNP sample was mixed with the LEAP assay reaction buffer (50 mM Tris pH 7.5, 50 mM KCl, 5 mM MgCl₂, 0.05% Tween-20, 0.2 mM dNTPs, 1 mM DTT, 0.4 μM 3′ RACE adapter (5′ - GCG AGC ACA GAA TTA ATA CGA CTC ACT ATA GGT TTT TTT TTT TTV N -3′), 40 units of RNase inhibitor (Enzynomics), total reaction volume: 50 μl) and incubated at 37 °C for 1 h. One microliter of LEAP cDNA products are subsequently amplified using 0.4 μM of L1 LEAP primer with the Phusion High-Fidelity polymerase kit (Thermo Fisher Scientific). PCR amplicons were separated and visualized in EtBr-stained 2% agarose gel.

**eIF3 PAR-CLIP analysis.** We utilized previously published PAR-CLIP data for eIF3a, b, d, and g[38]. Briefly, the authors immunoprecipitated eIF3b from 4-thiouridine-and-UV-treated 293 T cells to capture the eIF3-RNA complex. After high-salt washing and RNase digestion, they separated individual eIF3-RNA complexes through denaturing gel electrophoresis. eIF3a, b, d, and g were identified from four separate bands using mass-spectrometry and the interacting RNAs were purified and sequenced. Although three replicates were generated for each protein, only the first replicate was used for each. After retrieving the raw sequence files from NCBI (accession code: GSE65004), reads with low basecall qualities were excluded using the fastq_quality_filter from FASTX Toolkit (-q 25 -p 80; version 0.0.13.2; http://

hannonlab.cshl.edu/fastx_toolkit/). PCR duplicates were also excluded using fastx_collapser. Moreover, we excluded the reads that were shorter than 10-nt after trimming primer IDs and 3′ adapters from further analysis (cutadapt -a TGGAATTCTCGGGTGCCAAGG -u 12 -m 10; version 2.3[60];). The remaining reads were mapped to the L1Hs consensus sequence, wherein upto three mismatches were allowed using bowtie2[65] (--local –norc –score-min L,-18,2; version 2.2.4). For mean coverage analysis of 5′ UTR, ORF1, ORF2, and 3′ UTR, the number of reads that begin and end within each region were counted and the number was divided by the length of the corresponding region. The codes used for analyzing PAR-CLIP and mapping data are available from https://github.com/schanbaek/eif3_par_clip.

**Comparison analysis of species-specific m⁶A site.** To identify the species-specific full-length L1s in human, chimpanzee, and gorilla genome, we used BLAT-based and liftOver-based methods[47] with a computational approach. Only L1s of which insertion sites and two flanking regions are supported to be unique to the human, chimpanzee or gorilla genome by BLAT and liftOver were included in the further analyses. Then, we eliminated certain ambiguous elements containing gap sequence in the reference genome data and those that were less than 5.5 kb. The flanking sequences (2 kb, both upstream and downstream) of each species-specific L1 candidate were manually compared to the orthologous loci in human (GRCh37/hg19; Feb. 2009), chimpanzee (CSAC Pan_troglodytes-3.0/panTro5; May. 2016), gorilla (GSMRT3/gorGor5; May. 2016), and orangutan (Susie_PABv2/ponAbe3; Jan. 2018) genomes. The flanking sequences were used to identify the orthologous positions in the other genomes using BLAST-Like Alignment Tool (BLAT). We collected and retrieved the species-specific full-length L1s. We then classified the L1 subfamilies (L1Hs, L1PA2~L1PA5) using RepeatMasker utility[66]. Multiple sequence alignment of species-specific full-length L1s in each genome was performed using MUSCLE (MUltiple Sequence Comparison by Log- Expectation) under the default option[67]. The conserved sequence motifs at the three sites (A332, A495, and A600) were visualized using the program Weblogo[68]. Species-specific L1 loci are listed in Supplementary data 1–3.

**Statistical analysis.** GraphPad Prism 7.00 was used for statistical analysis. Two-sided student's t-test was used for unpaired data. Two-sided Kolmogorov–Smirnov test was used to assess the quantification of number and intensity of colocalizing puncta in Fig. 5g, h. For multiple comparisions, one-way ANOVA with Tukey's or Dunnett's multiple comparison test were used. P-values <0.05 were considered significant.

**Reporting summary.** Further information on research design is available in the Nature Research Reporting Summary linked to this article.

## Data availability

MeRIP-seq data and input total RNA-seq data are available on the NCBI Gene Expression Omnibus database under accession number GSE152328. All data supporting the findings of this study are available from the corresponding author upon reasonable request. Source data are provided with this paper.

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

## Acknowledgements
This work was supported by the grants IBS-R008-D1, NRF-2020R1A5A1018081, and NRF-2020R1A2C3011298 (to K.A.), funding from the Fellowship for Fundamental Academic Fields of SNU (to S.-Y.H.), and the BK21 plus fellowship. We are grateful to J. L. Garica-Perez, W. An, A. Roy-Engel, and A.J. Doucet for providing the L1 plasmids. Computational comparative genomic work was facilitated by the Compute Canada high-performance computing facilities. We gratefully acknowledge Center for Bio-Medical Engineering Core Facility at Dankook University for data analysis including computer server.

## Author contributions
S.-Y.H., S.L., J.C., and K.A. contributed to the conceptualization and designed the experiments. S.-Y.H., K.P., and H.K. performed the biochemical and cell biological experiments. H.J. and J.K.C. conducted and provided support for the MeRIP-seq analyses. S.M., W.T.,

P.L., and K.H. worked on the identification of gorilla-specific full-length L1s and performed comparative analyses among primate-specific L1 subfamilies. S.C.B. performed eIF3 PAR-CLIP seq analyses. S.L. performed RNA FISH and IFA. S.L., H.C.M., and H.Y.P. led the microscopic image analyses. B.K. and V.N.K. provided support during the generation of the MeRIP-seq cDNA libraries. Y.C. and V.N.K. contributed to the polysome profiling experiments. Y.-H.G. and H.-J.C. provided the cell lysates of hESCs. S.-Y.H., H.J., S.M., S.L., S.C.B., K.P., H.C.M., H.Y.P., K.H., and K.A. wrote the manuscript.

## Competing interests

The authors declare no competing interests.
