## [Peer Review File · Nature Communications]

REVIEWER COMMENTS

Reviewer #1 (Remarks to the Author):

In the manuscript entitled " L1 retrotransposons exploit RNA m6A modification as an evolutionary driving force" Hwang et al. explored the role of m6A in the life-cycle of human retrocompetent L1. With well-designed experiments, this study demonstrates that m6A 'writer' METTL3 facilitates L1 retrotransposition, whereas m6A 'eraser' ALKBH5 suppresses it, by promoting ORF1p translation. Furthermore, the authors demonstrate that such effect is mediated by a m6A cluster located in L1Hs 5'UTR. Elucidating L1 regulators is a challenging and growing area of molecular biology research. Specifically, in light of recent publications, understanding the role of N 6-methyladenosine in L1 regulation remains as an unanswered question. Thus, this work provides novel insights about this interesting question. The manuscript is informative and well written; specific comments are listed below.

- 1- Several of the main conclusions of the paper are based on cell-based L1 retrotransposition assays performed in HeLa cells transfected with pJJ101 L1 dn6 2.2 construct. As Klawitter and col. showed, L1 dn6 mobilization only reaches an efficiency of 20-30% in relation to the most frequently used L1Hs copy, L1.3 (PJJ101/L1.3), and consequently the number of Blastidicin-S resistant colonies obtained to calculate the retrotransposition rate can be quite low (Klawitter, 2016). Given that fact the interpretation and robustness of the results can change significantly, and the authors should show the wells and the graph containing absolute number of colonies obtained in each condition including the viability controls for figures 1B, 1C, 3D, 3F, 3G and 6C.
- 2- The authors conclude that m6A modification affects ORF1p levels partially based on immunoblot assays of lysates from pL1Hs-transfected HeLa cells (Fig. 1E, F, G and Fig.4C). Since small differences on transfection efficiency could lead to the misinterpretation of these results, the author should provide transfection efficiency controls (e.g. quantification of plasmid DNA in each condition by PCR or western-blot of another protein encoded by the plasmid).
- 3- Regarding Fig. 4E and 4F, the authors show that the 5'UTR m6A cluster recruits eIF3 for the efficient translation of L1 RNA and suggest that m6A modification could enable cap-independent translation. Considering that ORF1p translation has been previously shown to be strictly cap-dependent (Dmitriev, 2007), it would be very interesting if the authors could show eIF3 mechanism promoting ORF1p translation.
- 4- Related to RNP purification (Fig. 5B, 5C, and 5D), the authors claimed "We detected comparable levels of L1 RNA in the RNP fractions from pL1Hs- and pL1 m6A mut-expressing cells... Immunoblotting of the RNP fraction showed that the levels of RNP-associated ORF1p were diminished by L1 5' UTR m6A mutation (Fig. 5C)...We amplified LEAPproducts...m6A-abrogated L1 RNP produced cDNA at significantly lower levels than the wild-type L1 RNP did (Fig. 5D). These results reveal that the m6A cluster is necessary for L1 cDNA production, which suggests that the m6A cluster regulates ORF2p expression or its activity." However, the amount of RNA in each condition was not quantified (it was measured by non quantitative RT-PCR). Thus, the decreased amount of ORF1p as well as the reduced activity of the reverse transcriptase encoded by ORF2p in cells transfected with pL1m6A mut could be due to the reduced amount of purified RNPs in this condition. The authors' current results cannot unambiguously distinguish between these possibilities. To clarify this, the authors should measure the amount of L1 RNA by a real time RT-PCR. On the other hand, the m6A effect on ORF2p level should be directly measured as they did for ORF1p (western-blot or immunofluorescence detection of tagged ORF2p (e.g. pAD3TE1 transfected cells)).
- 5- Fig. 5F. In these panels, the immunofluorescence detection of ORF1p-T7 image depicting HeLa cells transfected with pAD3TE1 looks saturated, so the authors should provide images with non-saturated pixels in which the L1 cytoplasmic foci can be clearly distinguished as Doucet and col. showed using the same plasmid (Doucet et al. (2010)).
- 6- Liu et al (Science 2020) have shown that using a m6A demethylation system in mESCs targeting LINE-1 (dCas13b-FTO), demethylation increase the half lifetime of LINE1 RNA. The authors should comment this apparent discrepancy with their data.

Reviewer #2 (Remarks to the Author):

Hwang et al reported that m6A positively regulated L1 retrotransposon activity and ORF1p translation efficiency using reporter system. In addition, they identified several m6A sites at 5'URT of L1 full-length RNA that bound by eIF3 for cap-independent translation, and showed these m6A sites for functions in L1 evolutionary driving force. The experiments were well designed, and the results are very interesting.

1. All results were done by a reporter system that transfection of L1 retrotransposon reporter. As known and showed that L1 is high expressed in hESCs, the authors should showed the evidence that the de novo L1 integration sites were decreased upon METTL3 knockdown or increased upon ALKBH5 knockdown.
2. Fig 4B, why did pL1Hs show globally higher translation initiation compared to pL1 m6A mut.
3. Fig 4D, knockdown ALKBH5 also increased globally translation, suggesting ALKBH5 might suppresses mRNA translation. As known that GAPDH also contains m6A, whether it is also affected by ALKBH5. Could the authors explain the reason that chose GAPDH as control.
4. The authors showed that eIF3 bounds L1 RNA. It would be better the authors show that L1 retrotransposition activity is affected by eIF3.
5. The authors showed that m6A regulates ORF1p protein synthesis, whether ORF2p translation was also affected by m6A? If not, could the authors explain or predict the reason?
6. m6A reader YTHDF2 increases translation efficiency of its targeted m6A-modified mRNA through interaction with eIF3. Whether YTHDF2 binds m6A-modified L1 directly and recruits eIF3?

Reviewer #3 (Remarks to the Author):

Review of Hwang et al. entitled, "L1 retrotransposons exploit RNA m6A modification as an evolutionary driving force" for consideration at Nature Communications

Summary

The authors provide a well-designed set of experiments, which lead to the hypothesis that m6A modification of LINE-1 RNA enhances the L1 translational efficiency and retrotransposition in cultured human cell models. The paper is well written, the experiments are well designed, and the emergent hypothesis will be of great interest to the L1 retrotransposon community.

Below, I have noted some comments for the authors to consider when revising their paper for publication in Nature Communications.

Comments

(1) Line 21: the authors may wish to cite more recent primary papers detailing how host cell factors may inhibit or facilitate retrotransposition to help build their argument. For example, Miyoshi et al. recently published that PARP2 and RPA act to facilitate retrotransposition, which in turn, may lead to APOBEC3 recruitment to inhibit retrotransposition. Similarly work on ZAP, APOBEC3, SAMHD1, etc. could be cited, but I understand if there are space limitations.

(2) Line 47: the blast vector was originally made by John Goodier and was first used in Morrish et al, Nature Genetics, 2002.

- (3) Line 56: probably better to say ~4-fold rather than >4-fold.
- (4) Line 86: the expression of endogenous L1s in hES cells was first shown by Garcia-Perez et al, Human Molecular Genetics, 2007; please include reference.
- (5) Line 101: please comment on the peaks near the start of ORF2 (where is it specifically) and the 3'UTR in Figure 2C. The inclusion of the methylated clusters at the base level would be a welcome addition to the paper.
- (6) Line 126: When numbering L1 bases, please put the accession number of the sequence you are using for the comparison.
- (7) Line 132: A big question from the paper is: How do the authors know that the mutations of the A332T, A495T, and/or 600T do not affect steady state RNA levels? They use RT-PCR later in the paper to address this point. However, a Northern blot, would go a long way to convince me that the mutations minimally affect transcription/transcript levels and act primarily at the level of translation. Please see Larson et al., PLoS Biology, 2018 for similar experiments used as controls in reporter gene based experiments.
- (8) Line 148: the signal to noise ratio seems quite low for some experiments in Figure 3F. The authors report ratios to monitor changes in retrotransposition. The inclusion of raw retrotransposition/luciferase data that supports the ratios reported throughout the study would be a welcome addition to the manuscript.
- (9) Line 161 (and throughout): L1 is not alive; please consider "L1 replication cycle".
- (10) Line 166: "when" compared.
- (11) Figure 4C (and related figures): in general, the quantification of expression changes observed in Western blots, etc. would be a welcome addition to the manuscript.
- (12) Line 220: it seems that ref. 39 is not referred to until later in the manuscript; please check.

Point-by-point responses (NCOMMS-20-26953)

You will find our responses to each of your points and suggestions. The reviewers' comments are in *italics*.

Reviewer #1 (Remarks to the Author):

In the manuscript entitled " L1 retrotransposons exploit RNA m6A modification as an evolutionary driving force" Hwang et al. explored the role of m6A in the life-cycle of human retrocompetent L1. With well-designed experiments, this study demonstrates that m6A 'writer' METTL3 facilitates L1 retrotransposition, whereas m6A 'eraser' ALKBH5 suppresses it, by promoting ORF1p translation. Furthermore, the authors demonstrate that such effect is mediated by a m6A cluster located in LIHs 5'UTR. Elucidating L1 regulators is a challenging and growing area of molecular biology research. Specifically, in light of recent publications, understanding the role of N 6-methyladenosine in L1 regulation remains as an unanswered question. Thus, this work provides novel insights about this interesting question. The manuscript is informative and well written; specific comments are listed below.

1- Several of the main conclusions of the paper are based on cell-based L1 retrotransposition assays performed in HeLa cells transfected with pJJ101 L1 dn6 2.2 construct. As Klawitter and col. showed, L1 dn6 mobilization only reaches an efficiency of 20-30% in relation to the most frequently used LIHs copy, L1.3 (PJJ101/L1.3), and consequently the number of Blastidicin-S resistant colonies obtained to calculate the retrotransposition rate can be quite low (Klawitter, 2016). Given that fact the interpretation and robustness of the results can change significantly, and the authors should show the wells and the graph containing absolute number of colonies obtained in each condition including the viability controls for figures 1B, 1C, 3D, 3F, 3G and 6C.

Response: We agree. At the beginning of this study, we checked and compared the retrotransposition frequency of pJJ101 L1 dn6 2.2(pLIHs) and pJJ101 L1.3. Indeed, the retrotransposition efficiency of pLIHs was lower than that of pJJ101 L1.3. Nevertheless, the reason why we did not use pJJ101/L1.3 was that L1.3 contains 4 bp insertion mutation in 5' UTR. This insertion is very exceptional and is not observed in any other L1PA1 consensus sequences (Figure below). Additionally, SRAMP analysis tool gave a high m⁶A prediction score at the 4 bp insertion. This site could be the reason for the high frequency of pJJ101 L1.3. However, this unique property of L1.3 made us use the other L1 plasmid, pLIHs which can better represent the consensus sequence of L1PA1.

Following your suggestion, we attached pictures of the wells for all L1 blasticidin-S assay (Supplementary figure 1A, 1C, 1D, 5A, 5B, 5F, 5G, 11A), and revised all the graph to show the absolute number of colonies (Figure 1B, 1C, 1D, 3D, 3F, 3G, 6C). Also, we measured cell viability using the Annexin V assay in each condition of L1 blasticidin-S assay (Supplementary figure 1B, 1E, 5C, 5H, 11B).

```

      651   660   670   680
      |-----|-----|-----|-----|
Ta1d  ACGCAGCTGGAGATCTGAGAACGGGCAGAC----TGC
L1rp  ACGCAGCTGGAGATCTGAGAACGGGCAGAC----TGC
L1.2  ACGCAGCTGGAGATCTGAGAACGGGCAGAC----TGC
L1.3  ACGCAGCTGGAGATCTGAGAACGGGCAGACAGACTGC
PA1   ACGCAGCTGGAGATCTGAGAACGGGCAGAC----TGC
L1dn6 ACGCAGCTGGAGATCTGAGAACGGGCAGAC----TGC
Consensus ACGCAGCTGGAGATCTGAGAACGGGCAGAC....TGC
```

Figure 1 for reviewers. Sequence comparison of L1PA1s in the 5' UTR 651-687 region.

2- The authors conclude that m6A modification affects ORF1p levels partially based on immunoblot assays of lysates from pL1Hs-transfected HeLa cells (Fig. 1E, F, G and Fig.4C). Since small differences on transfection efficiency could lead to the misinterpretation of these results, the author should provide transfection efficiency controls (e.g. quantification of plasmid DNA in each condition by PCR or western-blot of another protein encoded by the plasmid).

Response: As you suggested, the quantification of plasmid DNA by quantitative PCR are now included in Supplementary figure 2A, 2B and 7A.

3- Regarding Fig .4E and 4F, the authors show that the 5'UTR m6A cluster recruits eIF3 for the efficient translation of L1 RNA and suggest that m6A modification could enable cap-independent translation. Considering that ORF1p translation has been previously shown to be strictly cap-dependent (Dmitriev, 2007), it would be very interesting if the authors could show eIF3 mechanism promoting ORF1p translation.

Response: Thank you for your suggestion. We have investigated whether eIF3 regulates ORF1p translation. However, there were several limitations to perform experiments with the ectopic expression of L1 plasmids in eIF3 knockdown cells. Firstly, eIF3 knockdown ceases cell growth¹. Secondly, eIF3 knockdown makes cells vulnerable to antibiotics (inhibiting protein synthesis) necessary for the selection of plasmid-transfected cells. These affected the transfection efficiency of L1 plasmids and made protein quantification impossible. Instead, we tested endogenous L1 ORF1p expression in siRNA-treated PA-1 cells. As expected, ORF1p translation was dependent on eIF3 expression (Supplementary figure 8C).

Additionally, we tested the cap-dependency of L1 ORF1p translation using heat shock stress. Heat shock stress suppresses most cap-dependent translation², but 5' UTR m⁶A of HSP70 mRNA was reported to enable efficient protein synthesis via cap-independent translation^{3,4}. However, the translation of HSP70 mRNA also requires 5' end m⁷GpppG -capping⁵, which suggests that cap-independent translation by 5' UTR m⁶A could be turned on during certain stress condition. Therefore, we speculated that L1 ORF1p translation might be regulated in similar way of HSP70. Indeed, through immunoblot assays we revealed that heat shock stress induced L1 ORF1p synthesis in pL1Hs-expressing HeLa cells, but not in pL1 m⁶A mut-expressing HeLa cells (Figure below). This result indicates that heat shock upregulates 5' UTR m⁶A of L1 RNA not only of HSP70 mRNA, also suggests that eIF3-mediated cap-independent translation is involved in the regulation of L1 ORF1p translation.

Figure 2 for reviewers. Heat shock stress promotes L1 ORF1p translation. pL1-transfected cells were incubated at 42°C for 1 h, and then were harvested at 9 h post heat shock stress. Cell lysates were subjected to western blot assay.

4- Related to RNP purification (Fig. 5B, 5C, and 5D), the authors claimed “We detected comparable levels of L1 RNA in the RNP fractions from pL1Hs- and pL1 m6A mut-expressing cells... Immunoblotting of the RNP fraction showed that the levels of RNP-associated ORF1p were diminished by L1 5' UTR m6A mutation (Fig. 5C)... We amplified LEAPproducts... m6A-abrogated L1 RNP produced cDNA at significantly lower levels than the wild-type L1 RNP did (Fig. 5D). These results reveal that the m6A cluster is necessary for L1 cDNA production, which suggests that the m6A cluster regulates ORF2p expression or its activity.” However, the amount of RNA in each condition was not quantified (it was measured by non quantitative RT-PCR). Thus, the decreased amount of ORF1p as well as the reduced activity of the reverse transcriptase encoded by ORF2p in cells transfected with pL1m6A mut could be due to the reduced amount of purified RNPs in this condition. The authors' current results cannot unambiguously distinguish between these possibilities. To clarify this, the authors should measure the amount of L1 RNA by a real time RT-PCR. On the other hand, the m6A effect on ORF2p level should be directly measured as they did for ORF1p (western-blot or immunofluorescence detection of tagged ORF2p (e.g. pAD3TE1 transfected cells)).

Response: We agree with your opinion and idea. According to your suggestion, we measured the amount of L1 RNA from purified RNPs using quantitative real-time PCR (Supplementary figure 9A). As a result, we reached the same conclusion. Also, we tried to detect L1 ORF2p levels using pAD3TE1-transfected cells. Through western blot analysis, we obtained similar results of 5' UTR m⁶A-mediated ORF1p expression in pAD3TE1 transfected cells (Figure below). Unlike L1 ORF1p, it has been hypothesized that as few as one molecule of ORF2p is translated per L1 RNA molecule⁶. This has made it difficult to detect ORF2p expression even using ectopic expression systems. Consistently, we confirmed extremely low expression levels of TAP-tagged ORF2p. Although the low expression level of L1 ORF2p makes quantitative comparison difficult, L1 ORF2p translation seems to be unaffected by m⁶A.

Figure 3 for reviewers. A schematic of pAD3TE1 L1 plasmid used in this study (left). Immunoblot assay of HeLa cells expressing pAD3TE1 L1Hs or pAD3TE1 L1 m⁶A mut (right).

5- Fig. 5F. In these panels, the immunofluorescence detection of ORF1p-T7 image depicting HeLa cells transfected with pAD3TE1 looks saturated, so the authors should provide images with non-saturated pixels in which the L1 cytoplasmic foci can be clearly distinguished as Doucet and col. showed using the same plasmid (Doucet et al. (2010)).

Response: As requested, we have attached the non-saturated version of Figure 5F in supplementary figure 9B. We were aware of this concern, but the images were needed to be adjusted for the weak intensity of pAD3TE1 L1 m⁶A mut. Additionally, we attached another cell image in the same experiment (Figure below).

Figure 4 for reviewers. Comparison of immunofluorescence and RNA FISH images depicting HeLa cells transfected with pAD3TE1. Figure 5F (left), unsaturated image of figure 5F (middle), and another cell image (right) were displayed.

6- Liu et al (Science 2020) have shown that using a m6A demethylation system in mESCs targeting LINE-1 (dCas13b-FTO), demethylation increase the half lifetime of LINE1 RNA. The authors should comment this apparent discrepancy with their data.

Response: Thank you for your critical comment. Most L1s can no longer mobilize because of 5' truncation, inversion, or point mutations^{7,8}. Only 0.01% or less of the entire L1s (80-100 copies) are potentially active in human genome⁹. In this perspective, Liu et al. observed the role of m⁶A in genome-wide L1s, which are mostly not full-length. However, our study focused on the functions of m⁶A in the replication cycle of retrotransposition-competent and full-length L1s. Thus, 5' UTR m⁶A cluster-mediated L1 regulation might not be observed in the study of Liu et al. We have addressed this issue in the revised discussion section.

Reviewer #2 (Remarks to the Author):

Hwang et al reported that m⁶A positively regulated L1 retrotransposon activity and ORF1p translation efficiency using reporter system. In addition, they identified several m⁶A sites at 5'URT of L1 full-length RNA that bound by eIF3 for cap-independent translation, and showed these m⁶A sites for functions in L1 evolutionary driving force. The experiments were well designed, and the results are very interesting.

1. All results were done by a reporter system that transfection of L1 retrotransposon reporter. As known and showed that L1 is high expressed in hESCs, the authors should showed the evidence that the *de novo* L1 integration sites were decreased upon METTL3 knockdown or increased upon ALKBH5 knockdown.

Response: As you suggested, we have investigated whether *de novo* insertions of endogenous L1 are regulated by METTL3 or ALKBH5. Since m⁶A metabolism is known to affect cell proliferation and self-renewal in hESCs¹⁰, we believe that it is difficult to perform such an experiment depleting essential m⁶A enzymes (e.g., METTL3 or ALKBH5) in hESCs. Therefore, we used another cell line, PA-1, which also highly expresses endogenous L1. In our study, we confirmed that m⁶A enzymes regulate ORF1p synthesis and its translation efficiency in PA-1 cells (Figure 4C, D). To measure *de novo* L1 integration, we generated METTL3 and ALKBH5-depleted PA-1 cell lines through shRNA lentiviral transduction (Figure below). Then we cultivated and cryopreserved these cells in passage 1, 6, and 12. These frozen cell stocks were used for genomic DNA (gDNA) extraction. Through gDNA-qPCR, we measured the amount of L1 DNA in gDNA using L1 ORF1-specific primers and 3' UTR-specific primers. Remarkably, depletion of ALKBH5 increased the enrichment level of L1 3' UTR, but not that of L1 ORF1 (Figure below). This result can be attributed to the frequent 5'-truncation of the L1 cDNA during insertion¹¹. However, METTL3 depletion, which decreases m⁶A modification, did not affect the enrichment of genomic L1. This result may be due to the tendency of L1 RNA to remain demethylated in the basal status of PA-1 cells. As results, changes in the levels of genomic L1s were not observed in control cells as well as METTL3-depleted cells. Despite these results, qPCR detection of genomic L1s could not provide the information of L1 integration sites. Therefore, to clarify the evidence of *de novo* L1 insertion, retrotransposon capture sequencing (RC-seq) would be necessary.

Figure 5 for reviewers. Scheme of generation and cultivation of METTL3- or ALKBH5-depleted cells (left, top). Immunoblot assay showing knockdown efficiency of target proteins in shRNA-transduced PA-1 cells (left, bottom). Quantification of genomic L1s by qPCR using METTL3-or ALKBH5-depleted cells (right). Cell passing numbers are indicated.

2. Fig 4B, why did pL1Hs show globally higher translation initiation compared to pL1 m6A mut.

Response: We conducted the polysome profiling experiments using cycloheximide. Since cycloheximide freeze translating ribosomes, but not initiating ribosome subunits¹², it could induce signal noise in 40S-80S fraction of polysome sedimentation. Also, if the translation is affected by some factor, the overall pattern of polysome profiling will be greatly changed. These two reasons made it difficult for us to conclude that pL1Hs-expressing cells exhibit higher translation initiation compared to pL1 m⁶A mut-expressing cells. Furthermore, we provided *GAPDH* mRNA quantification data including 3-5 fractions which correspond to 40S, 60S, and 80S (Figure below). In this result, we did not observe any significant changes in 3-5 fractions.

Figure 6 for reviewers. Relative mRNA levels of polysome-bound GAPDH. 40S, 60S, and 80S fractions were added to the supplementary figure 7B. Statistical significance was not observed. (n = 3 independent samples, error = s.e.m., unpaired Student's t test; **p < 0.01, *p < 0.05)

3. Fig 4D, knockdown *ALKBH5* also increased globally translation, suggesting *ALKBH5* might suppresses mRNA translation. As known that *GAPDH* also contains m6A, whether it is also affected by *ALKBH5*. Could the authors explain the reason that chose *GAPDH* as control.

Response: Since *GAPDH* does not contain m⁶A sites, many studies use *GAPDH* mRNA as an m⁶A modification negative control^{13,14}. In addition, we confirmed that *GAPDH* is suitable negative control through MeRIP-qPCR (Figure below). Additionally, abundant translation of *GAPDH* led us to choose *GAPDH* mRNA as control in polysome profiling

Figure 7 for reviewers. MeRIP-qPCR analysis of mRNA from HeLa cells. (n = 4 independent samples, error = s.d.)

4. The authors showed that eIF3 binds L1 RNA. It would be better the authors show that L1 retrotransposition activity is affected by eIF3.

Response: Following your suggestion, we have investigated whether eIF3 affects L1 mobility. However, as eIF3 is an essential translation initiation factor, its knockdown suppressed cell viability when we used translation-inhibiting antibiotics such as hygromycin B and blasticidin S. Thus, instead of L1-blasticidin assay, we conducted L1-luciferase assay in eIF3-depleted cells. The transfection efficiency of L1 plasmids was normalized to the luminescence of *renilla* luciferase (Supplementary figure 1F). As expected, eIF3 knockdown restrained L1 mobility, which indicates L1 retrotransposition is dependent on eIF3 (Supplementary figure 8D).

5. The authors showed that m6A regulates ORF1p protein synthesis, whether ORF2p translation was also affected by m6A? If not, could the authors explain or predict the reason?

Response: Thank you for your suggestion. Since as few as one molecule of ORF2p is translated per L1 RNA molecule⁶, ORF2p has been notoriously difficult to detect even using overexpression of L1 plasmid^{15,16}. As an alternative and as reviewer #1 has suggested, we tried to detect TAP-tagged L1 ORF2p levels by using pAD3TE1-transfected cells (Figure 3 for reviewers). Using TAP-tag antibody, we could detect ORF2p expression despite its extremely low expression levels. However, 5' UTR m⁶A did not seem to affect L1 ORF2p synthesis (Figure 3 for reviewers). This observation led us to hypothesize that 5' UTR m⁶A cluster may play role in ORF2p activity rather than in ORF2p translation.

6. m6A reader YTHDF2 increases translation efficiency of its targeted m6A-modified mRNA through interaction with eIF3. Whether YTHDF2 binds m6A-modified L1 directly and recruits eIF3?

Response: According to your suggestion, we have investigated the role of YTHDF1 and 2 in the recognition of L1 5' UTR m⁶A. Though eIF3 was reported to interact with YTHDF1¹⁷, it has remained unclear whether YTHDF2 binds to eIF3. First, we examined the interaction between YTHDF1/2 and eIF3. Through co-immunoprecipitation assay, we observed that both YTHDF1 and 2 are associated with eIF3 (Figure below). Furthermore, we have designed RNA-immunoprecipitation of YTHDFs in pL1Hs- or pL1 m⁶A mut-expressing HeLa cells. Remarkably, L1 RNA was detected in both eluates of YTHDF1 and 2 immunoprecipitation (Supplementary Fig 8E, F). However, 5' UTR m⁶A mutation did not impair the interaction between YTHDFs and L1 RNA (Supplementary Fig. 8F). Considering that YTHDFs mainly recognize m⁶A in 3' UTR, YTHDFs may not have a preference to the 5' UTR m⁶A cluster of L1 RNA.

Figure 8 for reviewers. Co-immunoprecipitation assay of HA-YTHDF1 and 2 using HA-antibody.

Reviewer #3 (Remarks to the Author):

Review of Hwang et al. entitled, “L1 retrotransposons exploit RNA m⁶A modification as an evolutionary driving force” for consideration at Nature Communications

Summary

The authors provide a well-designed set of experiments, which lead to the hypothesis that m⁶A modification of LINE-1 RNA enhances the L1 translational efficiency and retrotransposition in cultured human cell models. The paper is well written, the experiments are well designed, and the emergent hypothesis will be of great interest to the L1 retrotransposon community.

Below, I have noted some comments for the authors to consider when revising their paper for publication in Nature Communications.

Comments

(1) Line 21: the authors may wish to cite more recent primary papers detailing how host cell factors may inhibit or facilitate retrotransposition to help build their argument. For example, Miyoshi et al. recently published that PARP2 and RPA act to facilitate retrotransposition, which in turn, may lead to APOBEC3 recruitment to inhibit retrotransposition. Similarly work on ZAP, APOBEC3, SAMHD1, etc. could be cited, but I understand if there are space limitations.

Response: We truly appreciate your suggestion. Despite being aware of the importance of recent research, we cannot afford space in the introduction section.

(2) Line 47: the blast vector was originally made by John Goodier and was first used in Morrish et al, Nature Genetics, 2002.

Response: As suggested, we revised the reference.

(3) Line 56: probably better to say ~4-fold rather than >4-fold.

Response: We strongly agree with your comment. We correct the sentence.

(4) Line 86: the expression of endogenous L1s in hES cells was first shown by Garcia-Perez et al, Human Molecular Genetics, 2007; please include reference.

Response: As suggested, we have now added the reference in our revised manuscript.

(5) Line 101: please comment on the peaks near the start of ORF2 (where is it specifically) and the 3'UTR in Figure 2C. The inclusion of the methylated clusters at the base level would be a welcome addition to the paper.

Response: Through m⁶A prediction analysis, it was found that the other regions rather than 5' UTR did not contain m⁶A putative motif (Supplementary table 1A). Upon recognizing that we lacked explanation, we have added a statement that you pointed out. Now it reads: “Given that the reads from L1s may yield false-positive results, we narrowed down and selected the peaks that are likely to contain m⁶A motifs from 18 peaks through the m⁶A prediction score algorithm (SRAMP). SRAMP analysis revealed that the 9 peaks found in the ORF1, ORF2 and 3' UTR regions do not contain m⁶A motifs, and that only the 6 peaks located at 5' UTR have potential m⁶A motifs (Supplementary Table 1A).”

(6) Line 126: *When numbering L1 bases, please put the accession number of the sequence you are using for the comparison.*

Response: We agree. The numbering of L1 bases was based on the L1PA1 consensus sequence from the study of Khan et al¹⁸. However, Khan et al. did not provide the accession number of the sequence, but only supplementary information. Therefore, we have added reference when numbering L1 bases.

(7) Line 132: *A big question from the paper is: How do the authors know that the mutations of the A332T, A495T, and/or 600T do not affect steady state RNA levels? They use RT-PCR later in the paper to address this point. However, a Northern blot, would go a long way to convince me that the mutations minimally affect transcription/transcript levels and act primarily at the level of translation. Please see Larson et al., PLoS Biology, 2018 for similar experiments used as controls in reporter gene based experiments.*

Response: We agree with your concern. We performed a northern blot assay of L1Hs and L1 m⁶A mutant RNA using *mblastI*-deletion L1 constructs. For the detection of L1 mRNA, we used the radioisotope-labelled L1 5' UTR probe (1-232 region). The assay revealed that the steady-state of the L1 RNA level was not affected by 5' UTR m⁶A cluster (Supplementary figure S6A).

(8) Line 148: *the signal to noise ratio seems quite low for some experiments in Figure 3F. The authors report ratios to monitor changes in retrotransposition. The inclusion of raw retrotransposition/luciferase data that supports the ratios reported throughout the study would be a welcome addition to the manuscript.*

Response: As suggested, we attached the source data of our study. Also, we provided pictures of the wells for all L1 blasticidin-S assay (Supplementary figure 1A, 1C, 1D, 5A, 5B, 5F, 5G, 11A), and revised all the graph to show the absolute number of colonies (Figure 1B, 1C, 1D, 3D, 3F, 3G, 6C).

(9) Line 161 (and throughout): *L1 is not alive; please consider “L1 replication cycle”.*

Response: We apologize for the error. We have corrected the word.

(10) Line 166: *“when” compared.*

Response: As you pointed out, we revised the sentence.

(11) Figure 4C (and related figures): *in general, the quantification of expression changes observed in Western blots, etc. would be a welcome addition to the manuscript.*

Response: As you suggested, we displayed the quantification of L1 ORF1p expression changes under the immunoblot figure.

(12) Line 220: *it seems that ref. 39 is not referred to until later in the manuscript; please check.*

Response: We appreciate the reviewer's close attention. We have corrected the reference.

- 1 Wagner, S., Herrmannová, A., Malík, R., Peclínovská, L. & Valášek, L. S. Functional and Biochemical Characterization of Human Eukaryotic Translation Initiation Factor 3 in Living Cells. *Molecular and Cellular Biology* **34**, 3041-3052, doi:10.1128/mcb.00663-14 (2014).
- 2 Holcik, M. & Sonenberg, N. Translational control in stress and apoptosis. *Nature Reviews Molecular Cell Biology* **6**, 318-327, doi:10.1038/nrm1618 (2005).
- 3 Meyer, K. D. *et al.* 5' UTR m(6)A Promotes Cap-Independent Translation. *Cell* **163**, 999-1010, doi:10.1016/j.cell.2015.10.012 (2015).
- 4 Zhou, J. *et al.* Dynamic m(6)A mRNA methylation directs translational control of heat shock response. *Nature* **526**, 591-594, doi:10.1038/nature15377 (2015).
- 5 Song, H. J., Gallie, D. R. & Duncan, R. F. m7GpppG cap dependence for efficient translation of Drosophila 70-kDa heat-shock-protein (Hsp70) mRNA. *Eur J Biochem* **232**, 778-788 (1995).
- 6 Alisch, R. S., Garcia-Perez, J. L., Muotri, A. R., Gage, F. H. & Moran, J. V. Unconventional translation of mammalian LINE-1 retrotransposons. *Genes & development* **20**, 210-224, doi:10.1101/gad.1380406 (2006).
- 7 Brouha, B. *et al.* Hot L1s account for the bulk of retrotransposition in the human population. *Proc Natl Acad Sci U S A* **100**, 5280-5285, doi:10.1073/pnas.0831042100 (2003).
- 8 Sassaman, D. M. *et al.* Many human L1 elements are capable of retrotransposition. *Nat Genet* **16**, 37-43, doi:10.1038/ng0597-37 (1997).
- 9 Beck, C. R. *et al.* LINE-1 retrotransposition activity in human genomes. *Cell* **141**, 1159-1170, doi:10.1016/j.cell.2010.05.021 (2010).
- 10 Wang, Y. *et al.* N6-methyladenosine modification destabilizes developmental regulators in embryonic stem cells. *Nat Cell Biol* **16**, 191-198, doi:10.1038/ncb2902 (2014).
- 11 Zingler, N. *et al.* Analysis of 5' junctions of human LINE-1 and Alu retrotransposons suggests an alternative model for 5'-end attachment requiring microhomology-mediated end-joining. *Genome Res* **15**, 780-789, doi:10.1101/gr.3421505 (2005).
- 12 Garreau de Loubresse, N. *et al.* Structural basis for the inhibition of the eukaryotic ribosome. *Nature* **513**, 517-522, doi:10.1038/nature13737 (2014).
- 13 Zheng, G. *et al.* ALKBH5 is a mammalian RNA demethylase that impacts RNA metabolism and mouse fertility. *Mol Cell* **49**, 18-29, doi:10.1016/j.molcel.2012.10.015 (2013).
- 14 Zeng, Y. *et al.* Refined RIP-seq protocol for epitranscriptome analysis with low input materials. *PLoS Biol* **16**, e2006092-e2006092, doi:10.1371/journal.pbio.2006092 (2018).
- 15 Ergun, S. *et al.* Cell type-specific expression of LINE-1 open reading frames 1 and 2 in fetal and adult human tissues. *J Biol Chem* **279**, 27753-27763, doi:10.1074/jbc.M312985200 (2004).
- 16 Goodier, J. L., Ostertag, E. M., Engleka, K. A., Seleme, M. C. & Kazazian, H. H., Jr. A potential role for the nucleolus in L1 retrotransposition. *Hum Mol Genet* **13**, 1041-1048, doi:10.1093/hmg/ddh118 (2004).
- 17 Wang, X. *et al.* N6-methyladenosine Modulates Messenger RNA Translation Efficiency. *Cell* **161**, 1388-1399, doi:<https://doi.org/10.1016/j.cell.2015.05.014> (2015).
- 18 Khan, H., Smit, A. & Boissinot, S. Molecular evolution and tempo of amplification of human LINE-1 retrotransposons since the origin of primates. *Genome Res* **16**, 78-87, doi:10.1101/gr.4001406 (2006).

REVIEWERS' COMMENTS

Reviewer #1 (Remarks to the Author):

The authors have addressed most of the main points carefully. I have no further comments at this time and approve publication.

Reviewer #2 (Remarks to the Author):

The authors have answers all of my questions. I suggested the authors include experimental results into the manuscript, not just for reviews only.

Reviewer #3 (Remarks to the Author):

The authors have done a very nice job addressing the previous criticisms. I recommend publication.